# TimeCapsule: Solving the Jigsaw Puzzle of Long-Term Time Series Forecasting with Compressed Predictive Representations

## Abstract

Recent deep learning models for Long-term Time Series Forecasting (LTSF) often emphasize complex, handcrafted designs and traditional methodologies, while simpler architectures like linear models or MLPs have occasionally outperformed these intricate solutions. In this paper, we revisit and organize the core ideas behind several key techniques, such as redundancy reduction and multi-scale modeling, which are frequently employed in advanced LTSF models. Our goal is to streamline these ideas for more efficient deep learning utilization. To this end, we introduce TimeCapsule, a model built around the principle of high-dimensional information compression that unifies these key ideas in a generalized yet simplified framework. Specifically, we model time series as a 3D tensor, incorporating temporal, variate, and level dimensions, and leverage mode production to capture multi-mode dependencies while achieving dimensionality compression. We propose an internal forecast within the compressed representation domain, supported by the Joint-Embedding Predictive Architecture (JEPA) to monitor the learning of predictive representations. Extensive experiments on challenging benchmarks demonstrate the versatility of our method, showing that TimeCapsule can achieve performance comparable to state-of-the-art models. More importantly, the structure of our model yields intriguing empirical findings, prompting a rethinking of approaches in this area.

## 1 Introduction

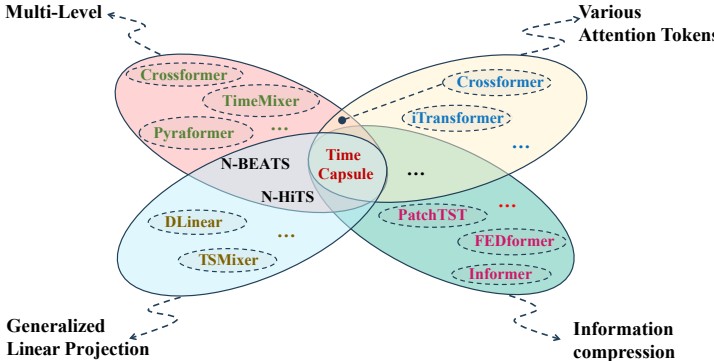

Figure 1: Categorization of advanced LTSF models into four groups based on their core techniques.

Multivariate Time Series (MvTS) data is one of the most ubiquitous forms of naturally generated data in the temporal physical world. Forecasting future events, whether short-term or long-term, based on these collected historical data, can support critical human activities, including finance (Sezer et al., 2020), traffic (Guo et al., 2020), and weather prediction (Karevan & Suykens, 2020). Moreover, it enables us to fundamentally explore the mechanisms underlying the world's operations (Bradley, 1999).

With the rapid advancement of deep learning models, Long-term Time Series Forecasting (LTSF) has recently gained prominence. Unlike short-term forecasting, LTSF has traditionally posed significant challenges for classical statistical and machine learning methods, such as VAR, ARIMA, and random forests (Toda & Phillips, 1994; Box & Pierce, 1970; Kane et al., 2014). A series of groundbreaking works (Zhou et al., 2021; Zeng et al., 2023; Nie et al., 2023; Liu et al., 2023) have been proposed to push the boundaries of this field, addressing limitations and advancing the community. Nevertheless, key questions remain about how best to improve existing models, leaving room for highlighting potential synergies among existing methods and combining their strengths. For instance, Informer suggests that the learned attention map of transformers should be sparse and can be distilled into smaller representations. Models like FiLM (Zhou et al., 2022a) and FEDformer (Zhou et al., 2022b) manipulate time series in compact frequency bases to capture key temporal correlations. These advancements raise a series of compelling questions: Can such enhancements be conveniently extended to other dimensions and more generalized transform domains? Is time series data, or the information it contains, inherently compressible? What's more, PatchTST applies the idea of patching to LTSF, enabling larger local receptive fields and greater efficiency. However, its reliance on predefined patch lengths introduces a rigidity that can be inflexible when dealing with varying input sequence lengths, thereby limiting its range of application. Additionally, this operation is not differentiable. Unlike image or text data, which contain explicit semantic information, splitting a time sequence into small chunks outside the training process may lead to unrecoverable information loss. While these methods individually have provided promising directions, they may overlook the combination of other critical principles in time series modeling and underestimate the capabilities of pure deep learning modules to represent these features.

Recent discussions in the community have also drawn attention to the surprisingly strong performance of linear models when compared to transformer-based architectures in LTSF tasks (Zeng et al., 2023). While transformers (Vaswani, 2017) have set milestones across domains like Natural Language Processing (NLP) (Wang et al., 2022) and Computer Vision (CV) (Yuan et al., 2021), their effectiveness in LTSF remains inconsistent, often falling short of simpler linear models (Zeng et al., 2023). In addressing this issue, iTransformer (Liu et al., 2023) offers valuable insights and experiments that transformers can excel when applied in the variate dimensions, but the ongoing debate over the performance of transformer-based versus linear-based models in LTSF is unresolved and whether transformers or linear models are more appropriate for LTSF remains open.

In this work, we argue that a deep learning framework for improving time series forecasting need not exclusively depend on either transformers or linear models. Instead, we propose that an effective architecture for LTSF should simply consist of two stages: predictive representation learning and generalized linear projection, in which the latter serves as an accurate predictor to learn generalized linear dependencies for the forecasting, while the former extracts abstract and informative representation from distinct data information to improve forecasters' generality. Therefore, we propose TimeCapsule, a novel model employing a Chaining Bits Back with Asymmetric Numeral Systems (BB-ANS) (Townsend et al., 2019) like-architecture, which adopts weak transformer-based blocks as the encoder, while MLP-based blocks as the decoder, striking a balance between powerful representation learning and computational simplicity. Technically, our model is driven by three key principles:

**Multi-level Modeling**: Multi-scale modeling stands out as an effective paradigm for improving LSTF performance (Ferreira et al., 2006). Existing methods incorporate multiresolution analysis by up/downsampling through moving average and convolutional pooling layers (Challu et al., 2023; Wang et al., 2024), or designing hierarchical structures (Liu et al., 2021; Zhang & Yan, 2023) to aggregate multi-scale features. Besides, time series decomposition is also a traditional and commonly used strategy to improve LTSF performance (Oreshkin et al., 2019; Zhou et al., 2022b; Wu et al., 2021). In avoidance of complexity, we propose adding an extra dimension called level to the original time series, allowing the model to learn multi-level features within the representation space. This approach generalizes multi-scale learning and time series decomposition, in the meanwhile, making the learning process model-independent.

**Multi-mode Dependency**: iTransformer has shown the benefits of leveraging correlations across dimensions other than time. However, focusing too heavily on non-temporal dimensions can risk neglecting important temporal dependencies, potentially degrading forecasting accuracy. To address this, we introduce a Mode-specific Multi-head Self-Attention (MoMSA) mechanism, leveraging

tensor-based mode product techniques to capture dependencies along and across multiple dimensions including temporal, variate, and level.

**Compressed Representation Forecasting**: To enable efficient information utilization, fast multi-mode attention computations, and better long-range history processing, we conclude that *compression* is all we need. To be specific, rather than focusing exclusively on sparse attention and redundancy reduction, we employ low-rank transforms to replace patching, reducing dimensionality ahead of the attention computation, thus leading to an economic way to employ transformers. This strategy compresses the representation space, enabling efficient computation and robust long-range forecasting. The compressed space can serve as the intermediate stage for learning forecasts, allowing us to map the representation into the future landscape and recover it back into the real temporal domain.

Our contributions can be summarized as follows:

- We propose simple yet effective generalizations to existing LTSF techniques, including a model-independent multi-level modeling strategy and a Mode-Specific Multi-head Self-Attention (MoMSA) mechanism, resulting in a versatile forecasting model capable of handling diverse data characteristics.
- We introduce JEPA into time series forecasting, making it a useful tool for monitoring and analyzing the process of predictive representation learning.
- Extensive experiments on real-world datasets demonstrate the superiority of our approach, identifying areas for further exploration.

## 2 RELATED WORK

**Recent Advancements in Long-term Time Series modeling** Given the advantage of capturing long-term dependencies in long sequence data, researchers have increasingly applied transformers to LTSF tasks (Wen et al., 2022). Most recently, several significant works have been proposed, either to improve the forecasting performance of transformers or to reduce their computational complexity. Additionally, some models that diverge from transformer architectures have also exhibited considerable promise in this area. Based on the strategies by which they achieve success, we can categorize these models into four groups, with potential overlaps, as presented in Fig. 1. The **first group** (e.g., Autoformer (Wu et al., 2021), N-BEATs (Oreshkin et al., 2019), Pyraformer (Liu et al., 2021), N-Hits (Challu et al., 2023), Crossformer (Zhang & Yan, 2023), TimeMixer (Wang et al., 2024)) focuses on multi-level modeling, which incorporates multiresolution/multi-scale analysis and series decomposition within the model. These techniques enable the model to learn both coarse and fine-grained features within time series, facilitating the capture of hierarchical temporal patterns. The **second group** (e.g., Informer (Zhou et al., 2021), FEDformer (Zhou et al., 2022b), PatchTST (Nie et al., 2023)), leverages information redundancy to filter and extract high-energy temporal correlations, or models time series in the form of temporal patches, avoiding the inclusion of finer, noisier information in the forecasting process, improving the model's robustness and prediction accuracy., These advancements lead to progressive improvements in both effectiveness and efficiency. The **third group** (e.g., Crossformer, iTransformer (Liu et al., 2023)) explores the impact of attention applied across various dimensions of time series, offering novel perspectives for time series correlation extraction. This group demonstrates that capturing dependencies across different modes can significantly enhance forecasting performance, by providing a richer understanding of temporal and variable relationships. The **fourth group** (e.g., DLinear (Zeng et al., 2023), N-BEATs, N-Hits, TsMixer (Chen et al., 2023), TimeMixer) emphasizes the importance of generalized linear dependency modeling and capitalizes on linear or MLP-based architectures to establish highly effective forecasters. This validates the crucial role of learning the appropriate coefficients to combine the captured base components of time series and the dependencies they contain.

These four groups highlight four key factors in answering how to model and forecast time series effectively from different aspects. However, each model typically focuses on constructing complex modules that address only parts of these factors. In contrast, our model integrates all of these interesting factors into a comprehensive yet streamlined design.

**Joint-Embedding Predictive Architecture** JEPA (LeCun, 2022) realizes representation learning by optimizing an energy function between predicted representations of inputs and targets. It doesn't

rely on explicit contrastive losses (Khosla et al., 2020) but instead creates compatible embeddings through prediction, enhancing the flexibility and efficiency of representation learning. While JEPA has demonstrated success in learning predictable representations in vision (Assran et al., 2023) and video tasks (Bardes et al., 2024), these domains often benefit from spatial locality and semantic coherence, which are different from the temporal dependencies and complex long-range patterns present in time series data. Such distinctions pose unique challenges when applying JEPA to LTSF. Although some recent works like LaT-PFN (Verdenius et al., 2024) and TS-JEPA (Girgis et al., 2024) have explored JEPA in time series tasks, their focus differs significantly from the predictive demands of LTSF. Specifically, the former employs JEPA to construct a time series foundation model, demonstrating that JEPA can reinforce the latent embedding space of time series learning and result in a superior zero-shot performance. In contrast, the latter, TS-JEPA, harnesses JEPA to facilitate the realization and enhancement of the effectiveness of semantic communication systems.

## 3 TIMECAPSULE

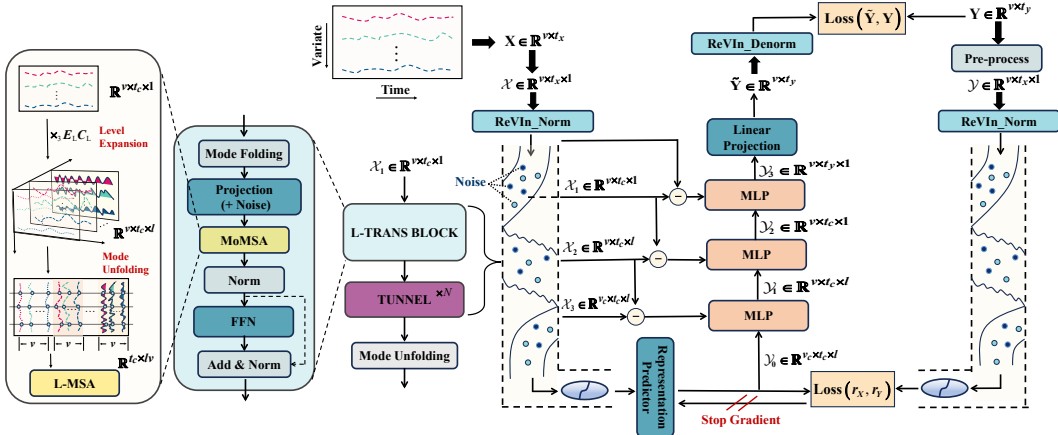

Figure 2: Overview of the TimeCapsule model. The original time series is transformed into a 3D representation by adding a level dimension, traverses through TransBlocks and tunnels (vanilla transformer blocks), and is then projected into the predictive space using JEPA. The compressed capsule is gradually recovered back into the real temporal domain.

Following the convention of Multivariate Time Series (MvTS) forecasting, we aim to predict the next $T_y$ steps of the time sequence, denoted as $Y = \{y_{t_x+1}, y_{t_x+2}, \cdots, y_{t_x+t_y}\} \in \mathbb{R}^{v \times t_y}$, based on the observed sequence $X = \{x_1, x_2, \cdots, x_{t_x}\} \in \mathbb{R}^{v \times t_x}$, where $v$ represents the number of variates and $t_x$ denotes the length of the input sequence. The key distinction in this work is that we treat MvTS as 3D tensor data, allowing the data itself to handle multi-scale modeling and series decomposition by introducing an additional dimension. Specifically, we extend the input from $X \in \mathbb{R}^{v \times t_x}$ to $\mathcal{X} \in \mathbb{R}^{v \times t_x \times 1}$. Throughout this paper, we use T, V, and L to represent the temporal dimension, variate dimension, and level dimension, respectively, with $t$, $v$, and $l$ indicating their corresponding lengths. We define $1 \leq t_c \ll t$, $1 \leq v_c \ll v$, and $l_c \geq 1$ as the compressed lengths of the respective dimensions. Additionally, $d$ denotes the size of the embedding space.

### 3.1 FEED FORWARD PROCESS

Generally, as depicted in Fig.2, TimeCapsule follows an asymmetric two-stage learning process: deep representation encoding and compressed information-based prediction. Internally, the encoder consists of three distinct stacks, each containing a transformer block followed by a series of tunnels. We place the time dimension T as the first block because it has a relatively long length, which should be compressed first to reduce the overall computational cost. The level expansion is placed second to enable multi-level learning in the representation space as efficiently as possible. Finally, we process the variable dimension. In contrast, the decoder is simpler, comprising just three MLP blocks. We

will first present an overview of the forward process, then go over a detailed explanation of how each of the key components is built and operates within this framework.

The encoding part can be formulized as

$$\mathcal{X} \leftarrow \mathrm{X} \in \mathbb{R}^{v \times t_x}, \ \ \mathcal{X}_0 = \mathrm{RevIn}(\mathcal{X}) \in \mathbb{R}^{v \times t_x \times 1} \tag{1}$$

$$\mathcal{X}_1 = \mathrm{Tunnel}(\text{T-TransBlock}(\mathcal{X}_0)) \in \mathbb{R}^{v \times t_c \times 1} \tag{2}$$

$$\mathcal{X}_2 = \mathrm{Tunnel}(\text{L-TransBlock}(\mathcal{X}_1)) \in \mathbb{R}^{v \times t_c \times l} \tag{3}$$

$$\mathcal{X}_3 = \mathrm{Tunnel}(\text{V-TransBlock}(\mathcal{X}_2)) \in \mathbb{R}^{v_c \times t_c \times l} \tag{4}$$

where $\mathrm{RevIn}(\cdot)$ denotes the reversible instance normalization proposed by (Kim et al., 2021), and Tunnels and TransBlocks are all transformer-based blocks. The prefix of TransBlock indicates the dimension along which the blocks are applied (T for temporal, L for level, and V for variate).

Next, the decoder operates in a reversed order of encoding process:

$$\mathcal{Y}_0 = \text{Repre\_Predictor}(\mathcal{X}_3) \in \mathbb{R}^{v_c \times t_c \times l} \tag{5}$$

$$\mathcal{Y}_1 = \mathrm{MLP}(\mathrm{Cat}(\mathcal{Y}_0, \mathcal{B}_3)) \in \mathbb{R}^{v \times t_c \times l} \tag{6}$$

$$\mathcal{Y}_2 = \mathrm{MLP}(\mathrm{Cat}(\mathcal{Y}_1, \mathcal{B}_2)) \in \mathbb{R}^{v \times t_c \times 1} \tag{7}$$

$$\mathcal{Y}_3 = \mathrm{MLP}(\mathrm{Cat}(\mathcal{Y}_2, \mathcal{B}_1)) \in \mathbb{R}^{v \times t_x \times 1} \tag{8}$$

where Repre-Predictor$(\cdot)$ is a single linear layer that projects the deep representation into the future landscape, and $\mathrm{Cat}(\cdot, \cdot)$ represents concatenation. $\mathcal{B}_1, \mathcal{B}_2, \mathcal{B}_3$ denote the residual information, which will be explained in detail later. The MLP block contains three linear layers with an intermediate GELU activation. Finally, we obtain the prediction result by another linear projection $\mathbb{R}^{t_y} \to \mathbb{R}^{t_y}$ and the inverse instance normalization.

$$\mathcal{Y} = \mathrm{Proj}(\mathcal{Y}_3) \in \mathbb{R}^{v \times t_y \times 1}, \ \ \mathcal{Y} \to \tilde{\mathrm{Y}} \in \mathbb{R}^{v \times t_y} \tag{9}$$

$$\mathrm{Y} = \mathrm{RevIn}(\tilde{\mathrm{Y}}) \in \mathbb{R}^{v \times t_y} \tag{10}$$

## 3.2 MAIN COMPONENTS

**Mode Specific Multi-head Self-Attention (MoMSA).** The MoMSA is designed to achieve two primary objectives: (a) to extend the ideas of crossformer (Zhang & Yan, 2023) and iTransformer (Liu et al., 2023) by forming MvTS tokens from a multi-mode view and abstracting dependencies along each mode; and (b) to maintain the same volume of information while shortening the length of each dimension, thereby reducing the overall computational cost of multi-mode self-attention.

To accomplish this, we introduce the mode-$k$ product (see Appendix B for the definition). For instance, given an MvTS tensor $\mathcal{A} \in \mathbb{R}^{v_a \times t_a \times l_a}$ and a transform factor $\mathrm{M} \in \mathbb{R}^{m \times t_a}$, MoMSA regarding to the temporal dimension operates as follows:

$$\hat{\mathcal{A}} = \mathcal{A} \times_2 \mathrm{M}, \ \ \mathrm{A}_1 = \text{T-MSA}(\hat{\mathrm{A}}_{(2)}) \in \mathbb{R}^{m \times v_a l_a} \tag{11}$$

where $\times_2$ denotes the mode-2 product, which represents a matrix multiplication along the second dimension of a 3D tensor. T-MSA denotes the vanilla multi-head self-attention (Vaswani, 2017) applied to the T (temporal) dimention, and $\hat{\mathrm{A}}_{(2)}$ is the mode-2 folding of $\hat{\mathcal{A}}$ (See Appendix B). It is notable that with the setting of $m \leq v_a$, the dimension can be compressed to an arbitrary length, resulting in a distilled attention map.

When applying this procedure to the real case, we shall obtain $\mathcal{X}_1 \in \mathbb{R}^{v \times t_c \times 1}$ from the first pipe as shown in the Fig.2. However, it is risky to make such a compression on the original MvTS information. To address this, we take a number of protective measures within our TransBlock. In particular, before applying MoMSA, we project the information into the embedding space $\mathbb{R}^{t_x \times d}$ and introduce Gaussian noise at the start of the block, which may improve robustness during compression (channel coding). Furthermore, we set the transform factor as the product of two matrices

$$\mathrm{M_T} = \mathrm{C_T} \mathrm{E_T}$$

where $\mathrm{E_T} \in \mathbb{R}^{t_e \times t_x}$ extends the dimension and $\mathrm{C_T} \in \mathbb{R}^{t_c \times t_e}$ does the compression, with $t_e \geq t_x \gg t_c$. By doing so, we enhance the information before compression, akin to the strategy used in

FEDformer, but within a generalized tranform domain. It is noteworthy that the transform factor is not necessarily invertible, as we allow lossy compression to help reduce information redundancy. In a nutshell, the deriviation of the temporal MoMSA can be formulized as

$$\text{MoMSA}(\mathcal{X}_0) = \text{T-MSA}((\text{Proj}(\mathcal{X}_0) + \mathcal{N}(0,1)) \times_2 \text{C}_\text{T}\text{E}_\text{T}) \tag{12}$$

where T is selected from the mode set $\{\text{T}, \text{L}, \text{V}\}$ to represent the T-TransBlock. Subsequently, additional transforms shall be applied in a sequential manner, whereby the symbol T shall be replaced with an alternative mode and the corresponding mode product shall be engaged. This process shall ultimately result in the generation of a compressed 3D representation, i.e., 'time capsule', at the end of the encoder.

Interestingly, our MoMSA can be viewed as a folded patch-wise self-attention mechanism that abstracts inter-correlations across different representation spaces. As illustrated in the leftmost example of Fig. 2, the two-dimensional information is obtained by folding the 3D tensor along the temporal dimension. This results in the formation of $l$ distinct groups, each representing a unique level, and containing $v$ variables. The attention token within the L-TransBlock is constituted by a combination of variables from different levels at each compressed timestamp. In this manner, traversing three TransBlocks with respect to various dimensions allows for the thorough capture of multi-mode dependencies. A comprehensive visual analysis can be found in the Appendix E.

**Residual Information Back.** One crucial aspect of our model to ensure more accurate predictions lies in leveraging complete but filtered history information. Thus it is required to compensate for the lost information during the decoding process. We transfer to the decoder the information calculated via residual subtractions instead of using the original $\mathcal{X}$, which may provide a shortcut for information retrieval.

$$\mathcal{B}_1 = (\mathcal{X}_0 - \mathcal{X}_1 \times_2 \text{C}_\text{T}\text{E}_\text{T}) \in \mathbb{R}^{v \times t_x \times 1} \tag{13}$$

$$\mathcal{B}_2 = (\mathcal{X}_1 - \mathcal{X}_2 \times_3 \text{C}_\text{L}\text{E}_\text{L}) \in \mathbb{R}^{v \times t_c \times 1} \tag{14}$$

$$\mathcal{B}_3 = (\mathcal{X}_2 - \mathcal{X}_3 \times_1 \text{C}_\text{V}\text{E}_\text{V}) \in \mathbb{R}^{v \times t_c \times l} \tag{15}$$

these residuals are subsequently used as outlined in Eq.6 to Eq.8.

**Representation Prediction with JEPA Loss.** In addition to the time-domain prediction, we introduce a predictor for the compressed representation, since we hypothesize a potential gap between the historical information (from the encoder) and future data (handled by the decoder). As described in Eq.5, we deploy a linear layer to make this projection. However, how to validate and ensure the meaningfulness of this inner prediction poses a question. For this reason, we employ the JEPA, which can not only introduce inductive bias into the time-variant process; but also allow an efficient contrastive loss between two representations outside of Euclidean space.

To implement JEPA, another encoder is required to convert the target into the same representation space as the input. In line with the strategies used in I-JEPA and V-JEPA, we continue to obtain the target encoder by applying the Exponential Moving Average (EMA) of the input encoder. However, due to the nature of LTSF, how to deal with the inconsistency of the sequence length between the input and output is problematic. we disentangle this problem by preprocessing the target sequence to match the input length as follows,

$$\text{Y} = \begin{cases} \text{Zero\_Padding}(\text{Y}, t_x - t_y) & \text{if } t_y < t_x \\ \text{EMA}(\{\text{Y}_{(k-1)t_x+1:kt_x}\}_{k=1}^{\lceil t_y//t_x \rceil}) & \text{if } t_y > t_x \end{cases} \tag{16}$$

i.e., in cases where $t_y < t_x$, we pad Y with zeros; and when $t_y$ is larger, we can reasonably consider the whole series as a carrier of information, and that the correlations between nearby timestamps are stronger. Under this assumption, we split the sequence Y into sub-sequences with length $t_x$ then applying EMA to these chunks to form integrated time information. This enables an efficient computation of the predictive representation loss:

$$\text{Loss}(\text{Enc}_x(\text{X}), \ sg(\text{Enc}_y(\text{Y})))$$

where $sg(\cdot)$ denotes the stop gradient operator. By incorporating this loss, we can quantify the distance between the learned representation and the target representation in the prediction space, and by default, we try adding this value to the final loss.

# 4 EXPERIMENT

To evaluate the performance and versatility of TimeCapsule, we conduct extensive experiments on ten diverse public datasets and compare the forecasting results against eight widely recognized forecasting models. More details on the datasets are provided in Appendix A.

## 4.1 FORECASTING RESULTS

**Datasets and Baselines.** We utilize datasets from various domains, including electricity (ETTh1, ETTh2, ETTm1, ETTm2, Electricity), environment (Weather), energy (Solar-Energy), transportation (PEMS04, Traffic) and health (ILI). The forecasting methods we compare against include iTransformer (Liu et al., 2023), TimeMixer (Wang et al., 2024), PatchTST (Nie et al., 2023), Crossformer (Zhang & Yan, 2023), DLinear (Zeng et al., 2023), TimesNet (Wu et al., 2022), FEDformer (Zhou et al., 2022b), and Informer (Zhou et al., 2021).

**Main Configurations.** All experiments are conducted on four NVIDIA 4090 GPUs with 24GB memory each. To align with JEPA, we use AdamW (Loshchilov, 2017) as the optimizer and Huber loss (Meyer, 2021) as the default loss function. Results are obtained using the random seed 2021. The batch size for each case is selected within the range of 32 to 128, and the learning rate is determined through a grid search between 1e-4 and 2e-3. TimeCapsule consists of 0 to 2 blocks of tunnels, with the compression dimension length chosen from the set $\{4, 8, 32\}$. To test the model's ability to utilize long-range historical data, we use a look back window of 512 for most datasets.

**Main Results.** Given the numerous benchmarks proposed in this area and a recent hidden bug discovered in the testing phase codes, making fair and trustworthy comparisons under a unified setting has proven challenging.

Table 1: Multivariate forecasting results. For TimeCapsule, the lookback length $T = 96$ and prediction lengths $S \in \{24, 36, 48, 60\}$ for ILI, $S \in \{96, 192, 336, 720\}$ and fixed lookback length $T = 512$ for others; For other models, lookback lengths are searched for the best performance as has been done by TFB (Qiu et al., 2024).

| Models | | **TimeCapsule** | | iTransformer (2023) | | TimeMixer (2024) | | PatchTST (2023) | | Crossformer (2023) | | DLinear (2023) | | TimesNet (2022) | | FEDformer (2022b) | | Informer (2021) | |
|---|---|---|---|---|---|---|---|---|---|---|---|---|---|---|---|---|---|---|---|
| Metric | | MSE | MAE | MSE | MAE | MSE | MAE | MSE | MAE | MSE | MAE | MSE | MAE | MSE | MAE | MSE | MAE | MSE | MAE |
| PEMS04 | 96 | **0.099** | **0.202** | 0.164 | 0.280 | 0.122 | 0.229 | 0.161 | 0.280 | 0.112 | 0.224 | 0.196 | 0.296 | 0.159 | 0.266 | 0.573 | 0.565 | 0.189 | 0.304 |
| | 192 | **0.117** | **0.222** | 0.216 | 0.316 | 0.141 | 0.239 | 0.178 | 0.290 | 0.134 | 0.236 | 0.213 | 0.310 | 0.179 | 0.282 | 0.655 | 0.624 | 0.229 | 0.335 |
| | 336 | **0.126** | **0.229** | 0.189 | 0.288 | 0.153 | 0.254 | 0.193 | 0.302 | 0.190 | 0.286 | 0.235 | 0.327 | 0.169 | 0.269 | 1.365 | 0.920 | 0.217 | 0.323 |
| | 720 | **0.137** | **0.239** | 0.251 | 0.351 | 0.174 | 0.276 | 0.233 | 0.338 | 0.235 | 0.331 | 0.327 | 0.395 | 0.187 | 0.286 | 0.873 | 0.728 | 0.310 | 0.391 |
| Weather | 96 | **0.141** | **0.186** | 0.159 | 0.208 | 0.147 | 0.198 | 0.149 | 0.196 | 0.146 | 0.212 | 0.170 | 0.230 | 0.170 | 0.219 | 0.223 | 0.292 | 0.218 | 0.255 |
| | 192 | **0.187** | **0.232** | 0.200 | 0.248 | 0.192 | 0.243 | 0.193 | 0.240 | 0.195 | 0.261 | 0.212 | 0.267 | 0.222 | 0.264 | 0.252 | 0.322 | 0.269 | 0.306 |
| | 336 | **0.239** | **0.272** | 0.253 | 0.289 | 0.247 | 0.284 | 0.244 | 0.281 | 0.268 | 0.325 | 0.257 | 0.305 | 0.293 | 0.310 | 0.327 | 0.371 | 0.320 | 0.340 |
| | 720 | **0.309** | **0.323** | 0.321 | 0.338 | 0.318 | 0.330 | 0.314 | 0.332 | 0.330 | 0.380 | 0.318 | 0.356 | 0.360 | 0.355 | 0.424 | 0.419 | 0.392 | 0.390 |
| Traffic | 96 | 0.361 | 0.246 | 0.363 | 0.265 | 0.466 | 0.294 | 0.379 | 0.270 | 0.514 | 0.282 | 0.410 | 0.282 | 0.600 | 0.313 | 0.593 | 0.365 | 0.664 | 0.371 |
| | 192 | **0.383** | **0.257** | 0.385 | 0.273 | 0.508 | 0.299 | 0.394 | 0.277 | 0.501 | 0.273 | 0.423 | 0.288 | 0.619 | 0.328 | 0.614 | 0.375 | 0.724 | 0.396 |
| | 336 | **0.393** | **0.262** | 0.396 | 0.277 | 0.526 | 0.309 | 0.402 | 0.280 | 0.507 | 0.278 | 0.436 | 0.296 | 0.627 | 0.330 | 0.609 | 0.373 | 0.796 | 0.435 |
| | 720 | **0.430** | **0.282** | 0.445 | 0.312 | 0.554 | 0.322 | 0.442 | 0.302 | 0.571 | 0.301 | 0.466 | 0.315 | 0.659 | 0.342 | 0.646 | 0.394 | 0.823 | 0.453 |
| Electricity | 96 | **0.125** | **0.218** | 0.138 | 0.237 | 0.131 | 0.224 | 0.133 | 0.233 | 0.135 | 0.237 | 0.140 | 0.237 | 0.164 | 0.267 | 0.186 | 0.302 | 0.214 | 0.321 |
| | 192 | **0.146** | **0.238** | 0.157 | 0.256 | 0.151 | 0.242 | 0.150 | 0.248 | 0.160 | 0.262 | 0.154 | 0.250 | 0.180 | 0.280 | 0.201 | 0.315 | 0.245 | 0.350 |
| | 336 | **0.161** | **0.254** | 0.167 | 0.264 | 0.169 | 0.260 | 0.168 | 0.267 | 0.182 | 0.282 | 0.169 | 0.268 | 0.190 | 0.292 | 0.218 | 0.330 | 0.294 | 0.393 |
| | 720 | **0.194** | **0.285** | 0.194 | 0.286 | 0.200 | 0.286 | 0.202 | 0.295 | 0.246 | 0.337 | 0.204 | 0.301 | 0.209 | 0.307 | 0.241 | 0.350 | 0.306 | 0.393 |
| ILI | 24 | **1.675** | **0.793** | 1.783 | 0.846 | 1.807 | 0.820 | 1.840 | 0.835 | 2.981 | 1.096 | 2.208 | 1.031 | 2.009 | 0.926 | 2.400 | 1.020 | 2.738 | 1.151 |
| | 36 | **1.725** | **0.831** | 1.746 | 0.860 | 1.896 | 0.927 | 1.724 | 0.845 | 3.295 | 1.162 | 2.032 | 0.981 | 2.552 | 0.997 | 2.410 | 1.005 | 2.890 | 1.145 |
| | 48 | **1.690** | **0.839** | 1.716 | 0.898 | 1.753 | 0.866 | 1.762 | 0.863 | 3.586 | 1.230 | 2.209 | 1.063 | 1.956 | 0.919 | 2.592 | 1.033 | 2.742 | 1.136 |
| | 60 | **1.775** | **0.863** | 1.960 | 0.977 | 1.828 | 0.930 | 1.752 | 0.894 | 3.693 | 1.256 | 2.292 | 1.086 | 2.178 | 0.962 | 2.539 | 1.070 | 2.825 | 1.139 |
| Solar | 96 | 0.173 | 0.229 | 0.188 | 0.242 | 0.178 | 0.231 | 0.190 | 0.273 | 0.166 | 0.230 | 0.216 | 0.287 | 0.285 | 0.330 | 0.509 | 0.530 | 0.338 | 0.373 |
| | 192 | **0.188** | **0.242** | 0.193 | 0.258 | 0.209 | 0.273 | 0.204 | 0.302 | 0.214 | 0.251 | 0.244 | 0.305 | 0.309 | 0.342 | 0.474 | 0.500 | 0.375 | 0.391 |
| | 336 | **0.194** | **0.248** | 0.195 | 0.259 | 0.209 | 0.259 | 0.212 | 0.293 | 0.203 | 0.260 | 0.263 | 0.319 | 0.338 | 0.365 | 0.338 | 0.439 | 0.417 | 0.416 |
| | 720 | **0.204** | **0.254** | 0.223 | 0.281 | 0.246 | 0.284 | 0.221 | 0.310 | 0.735 | 0.721 | 0.264 | 0.324 | 0.346 | 0.355 | 0.365 | 0.459 | 0.390 | 0.407 |
| ETTm2 | 96 | **0.161** | **0.249** | 0.175 | 0.266 | 0.172 | 0.265 | 0.165 | 0.254 | 0.263 | 0.359 | 0.164 | 0.255 | 0.190 | 0.266 | 0.219 | 0.306 | 0.216 | 0.302 |
| | 192 | **0.216** | **0.289** | 0.242 | 0.312 | 0.236 | 0.304 | 0.221 | 0.292 | 0.361 | 0.425 | 0.224 | 0.304 | 0.251 | 0.308 | 0.294 | 0.357 | 0.324 | 0.367 |
| | 336 | **0.269** | **0.324** | 0.282 | 0.340 | 0.273 | 0.329 | 0.275 | 0.325 | 0.469 | 0.496 | 0.277 | 0.337 | 0.362 | 0.350 | 0.362 | 0.401 | 0.424 | 0.429 |
| | 720 | **0.344** | **0.373** | 0.378 | 0.398 | 0.366 | 0.393 | 0.360 | 0.380 | 1.263 | 0.857 | 0.371 | 0.401 | 0.414 | 0.403 | 0.459 | 0.450 | 0.581 | 0.500 |
| ETTm1 | 96 | **0.284** | **0.340** | 0.300 | 0.353 | 0.336 | 0.371 | 0.290 | 0.343 | 0.310 | 0.361 | 0.299 | 0.343 | 0.377 | 0.398 | 0.467 | 0.465 | 0.430 | 0.424 |
| | 192 | **0.327** | **0.367** | 0.345 | 0.382 | 0.370 | 0.389 | 0.329 | 0.368 | 0.363 | 0.402 | 0.334 | **0.364** | 0.405 | 0.411 | 0.610 | 0.524 | 0.550 | 0.479 |
| | 336 | **0.355** | **0.382** | 0.374 | 0.398 | 0.397 | 0.410 | 0.360 | 0.390 | 0.408 | 0.430 | 0.365 | 0.384 | 0.443 | 0.437 | 0.618 | 0.544 | 0.654 | 0.529 |
| | 720 | **0.415** | **0.415** | 0.429 | 0.430 | 0.463 | 0.446 | 0.416 | 0.422 | 0.777 | 0.637 | 0.418 | 0.415 | 0.495 | 0.464 | 0.615 | 0.551 | 0.714 | 0.578 |
| ETTh2 | 96 | **0.272** | **0.338** | 0.297 | 0.348 | 0.280 | 0.350 | 0.277 | 0.339 | 0.611 | 0.557 | 0.302 | 0.368 | 0.319 | 0.363 | 0.338 | 0.380 | 0.378 | 0.402 |
| | 192 | **0.334** | **0.379** | 0.371 | 0.403 | 0.351 | 0.390 | 0.345 | 0.381 | 0.810 | 0.651 | 0.405 | 0.433 | 0.411 | 0.416 | 0.415 | 0.428 | 0.462 | 0.449 |
| | 336 | 0.367 | 0.409 | 0.404 | 0.428 | 0.366 | 0.414 | 0.368 | 0.404 | 0.928 | 0.698 | 0.496 | 0.490 | 0.415 | 0.443 | 0.378 | 0.451 | 0.426 | 0.449 |
| | 720 | **0.381** | **0.421** | 0.424 | 0.444 | 0.433 | 0.455 | 0.397 | 0.432 | 1.094 | 0.775 | 0.766 | 0.622 | 0.429 | 0.445 | 0.479 | 0.485 | 0.401 | 0.449 |
| ETTh1 | 96 | **0.362** | **0.394** | 0.386 | 0.405 | 0.373 | 0.401 | 0.376 | 0.396 | 0.405 | 0.426 | 0.371 | 0.392 | 0.389 | 0.412 | 0.379 | 0.419 | 0.709 | 0.563 |
| | 192 | 0.401 | 0.418 | 0.424 | 0.440 | 0.415 | 0.425 | 0.399 | 0.416 | 0.413 | 0.442 | 0.404 | 0.413 | 0.440 | 0.443 | 0.419 | 0.443 | 0.724 | 0.570 |
| | 336 | 0.432 | 0.440 | 0.449 | 0.460 | 0.454 | 0.453 | 0.418 | 0.432 | 0.442 | 0.460 | 0.434 | 0.435 | 0.482 | 0.465 | 0.455 | 0.464 | 0.732 | 0.581 |
| | 720 | **0.438** | **0.458** | 0.495 | 0.487 | 0.501 | 0.481 | 0.450 | 0.469 | 0.550 | 0.539 | 0.469 | 0.489 | 0.525 | 0.501 | 0.474 | 0.488 | 0.760 | 0.616 |
| 1st Count | | **33** | **35** | 0 | 0 | 1 | 0 | 4 | 3 | 1 | 0 | 0 | 2 | 0 | 0 | 0 | 0 | 0 | 0 |
| 2nd Count | | 7 | 4 | 9 | 5 | 5 | 5 | 15 | 15 | 3 | 6 | 2 | 2 | 0 | 0 | 0 | 0 | 0 | 0 |

Fortunately, thanks to the contributions from TFB (Qiu et al., 2024), a comprehensive and reliable benchmark specifically designed for LTSF is now available, with results obtained through meticulous adjustments. To ensure objective comparisons, the experimental results reported in this section

are partially derived from their work, and it is noteworthy that our basic settings are consistent with those adopted in the TFB benchmark.

As shown in Table 4.1, where the best results are highlighted in bold and the second best in underlined, each baseline demonstrates distinct advantages across different scenarios. However, our model consistently achieves or approaches the best performance across all datasets and forecasting horizons. While improvements compared to the second-best results are often marginal, TimeCapsule excels particularly in very long-term forecasting. These observations underscore the effectiveness and flexibility of our model, which integrates four key techniques in time series modeling. A central argument we propose is that different strategies may respond to various datasets to differing extents, revealing unique underlying characteristics. Taking the spatio-temporal dataset PEMS04 for a case study, TimeCapsule shows considerable progress compared to the second-best models, Crossformer and TimeMixer, with an average reduction of 15.8% in MAE and 10% in MSE. Notably, aside from Crossformer and TimeMixer, which emphasize careful multi-scale modeling, the performances of all other forecasters significantly degrade. This implies that forecasting on PEMS04 heavily relies on multi-scale modeling, a capability that TimeCapsule successfully possesses. Besides, our model outperforms PatchTST, particularly on larger datasets, without employing patching techniques. This indicates that TimeCapsule can also effectively leverage long historical information through dimension compression and MLP utilization.

## 4.2 MODEL ANALYSIS

**Effects of residual information compensation.** Information compensation should be indispensable if it is reasonable to construct such a deep representation utilization strategy that adheres to the principle of lossy compression. To verify its effects, we compare model performance after removing the compensation and replacing it with the original information. The results, listed in Table 2, reveal a dramatic drop in performance without paying back the lost information. This phenomenon indicates that the compressed information cannot be independently recovered under our default settings. However, when comparing our results with those of FEDformer and Informer, which also focus on redundancy reduction, our model demonstrates competitive results, validating the efficacy of our compression strategy. Additionally, compared to using the original information, the improvements observed on datasets like ETTh2 and ETTm2 are marginal. However, as the prediction horizon increases, the benefits of residual information become more pronounced. This suggests that redundancy reduction plays a more critical role in very long-term forecasting. For shorter forecasting lengths, the available information is typically sufficient. In contrast, when the forecasting horizon approaches or exceeds the input length, more compact and precise information is necessary, making the forecaster more sensitive to the quality and purity of the information provided. Notably, some degradations persist even with complete original information since it is sometimes challenging for our model to retrieve useful parts amid redundancy, which underscores the importance of our residual information computation.

Table 2: Ablation on the information compensation design. We remove the residual information connection or replace $\mathcal{B}$ by the original information $\mathcal{X}$. The resulting performance variations are then presented in terms of mean squared error (MSE).

| Method | ETTh2 | | | | ETTm2 | | | | PEMS04 | | | |
|---|---|---|---|---|---|---|---|---|---|---|---|---|
| | 96 | 192 | 336 | 720 | 96 | 192 | 336 | 720 | 96 | 192 | 336 | 720 |
| Origin | **0.272** | **0.334** | **0.367** | **0.381** | **0.161** | **0.216** | **0.269** | **0.344** | **0.099** | **0.117** | **0.126** | **0.137** |
| Replace Info | 0.276 | 0.347 | 0.374 | 0.410 | 0.161 | 0.219 | 0.269 | 0.352 | 0.114 | 0.130 | 0.130 | 0.163 |
| w/o Info Back | 0.371 | 0.387 | 0.393 | 0.462 | 0.289 | 0.317 | 0.352 | 0.422 | 0.273 | 0.247 | 0.290 | 0.339 |

**Effects of JEPA and Predictive Representation.** As aforementioned, JEPA loss can be employed to keep track of the learning process of the predictive representation and contributes to the final loss in our default setting. However, by removing it from backpropagation, we find that the role of JEPA is subtle and varies by case. For instance, as illustrated in Fig. 4.2, which records the variation of JEPA loss on the weather and ETTm2 datasets, the curve consistently decreases even without explicit training. This phenomenon partially confirms the existence of a gap between historical compression and future forecasting. The difference in converged JEPA loss between models trained

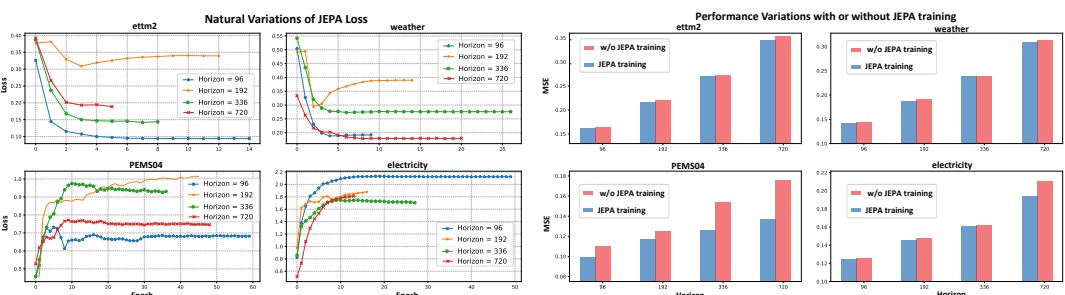

Figure 3: The left-hand figures illustrate the variation trend of JEPA loss in the absence of back-propagation, while the right-hand figures demonstrate the impact of incorporating JEPA loss as a training object on performance.

with and without JEPA roughly reflects the distance between two subsets of the metric space. In essence, the optimization direction indicated by JEPA loss minimization is promising.

On the other hand, this is completely a different case for PEMS04 and electricity datasets, which by contrast shows a growing trend in JEPA loss. We compare forecasting accuracy with their origin values to make it further, as shown on the right of Fig. 4.2, from which we can empirically conclude that in most cases, regularizing the optimization with the inductive bias introduced by JEPA is unlikely to cause harm. Incorporating JEPA loss into the training process can encourage the forecaster to converge to a favorable minimum before stepping into other directions. However, the effect of attaching such a forcement becomes negligible when the descent directions of two losses align or when have reached a flat energy plane. This finding may provide greater flexibility in leveraging JEPA for optimization guidance in predictive representation learning.

Table 3: Investigation on compression dimensions. Dimension set contains lengths of compression dimension of {T(temporal), V(variate), L(level)}. MSE and MAE are both reported by averaging those from all prediction horizons, and ╱ denotes performance decline while — means that there are almost no changes in performance.

| Dimension Set | PEMS04 | | Weather | | Electricity | | Traffic | |
|---|---|---|---|---|---|---|---|---|
| | MSE (avg) | MAE (avg) | MSE (avg) | MAE (avg) | MSE (avg) | MAE (avg) | MSE (avg) | MAE (avg) |
| (4, 8, 4) | 0.120 | 0.223 | 0.219 | 0.253 | 0.157 | 0.249 | 0.392 | 0.262 |
| (4, 1, 4) | 0.136 | 0.236 | 0.220 | 0.255 | 0.158 | 0.251 | 0.404 | 0.277 |
| (1, 1, 1) | 0.156 | 0.257 | 0.220 | 0.254 | 0.159 | 0.250 | 0.410 | 0.285 |
| Average Variation (%) | ╱ 21.7 | ╱ 10.6 | – | – | – | – | ╱ 3.9 | ╱ 7.2 |

**Analysis of Asymmetric Structure.** We investigate the asymmetric structure of TimeCapsule by varying the compression dimensions to figure out which part—transformer-based encoder or MLP-based decoder—plays a more crucial role. By default, the compression dimensions are set to {4, 8, 4}. Then we test two alternative configurations: (a) {4, 1, 4}, which removes the level embedding effect, and (b) {1, 1, 1}, which reduces the entire model to a pure MLP. The results in Table 3 reveal that for datasets with higher sampling frequencies, such as PEMS04, multi-level modeling and capturing multi-mode dependencies show significant benefits. In contrast, for datasets like Weather and Electricity, a simpler MLP structure appears to be sufficient for long-term forecasting (see Appendix D.5 for results of more datasets). These observations, along with the comprehensive results in Table 4.1, suggest that designing a universal, efficient model for diverse datasets is challenging, often leading to inefficient module allocation. Furthermore, a forecaster with strong generalized linear modeling capacity can manage most cases effectively and is more likely the key principle for LTSF.

### 4.3 EFFICIENCY STUDY.

Despite TimeCapsule containing multiple transformer blocks along three dimensions, it maintains a reasonable computational complexity due to the use of dimensionality compression and MLPs. To confirm this, we conduct efficiency comparisons on two datasets, Weather and Traffic, analyzing

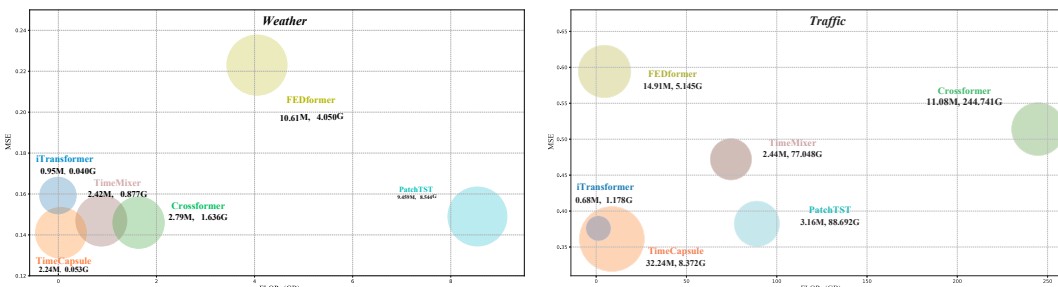

Figure 4: Efficiency comparisons in terms of FLOPs (GB) and parameter counts (MB) with the latest advanced models on the weather and traffic datasets. Statistics for each model are obtained under the default settings with same batch size.

both FLOPs and parameter counts. As shown in Fig. 4, our model demonstrates clear advantages in computation speed, though memory usage can sometimes be high. Nevertheless, a principal advantage of TimeCapsule is the effect-efficient balance attained by its asymmetric structure, which permits a substantial reduction in parameters without much sacrifice in performance.

## 5 CONCLUSION

We propose and empirically evaluate a generic model called TimeCapsule for long-term multivariate time series forecasting. We avoid incorporating explicit designs focused on the core characteristics of time series modeling, but only leverage the learning capacity of generic deep learning modules, complemented by simple strategies such as 3D tensor modeling and multi-mode transforms. By conceptualizing the forecasting process as an information compression task, we integrate JEPA to both guide and detect the learning of predictive representations. While our approach offers promising results, we believe there are even more effective ways to implement this framework. Additionally, extending this modeling approach to explore time series self-supervised learning, transfer learning, and other related time series tasks represents a promising step for future research.

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

# A DATASET DESCRIPTIONS

Ten widely-used public datasets are included in our experiment. ETTh1, ETTh2, ETTm1, and ETTm2 (Zhou et al., 2021) represent datasets recording hourly and 15-minute intervals of Electricity Transformer Temperature. The Weather dataset (Wu et al., 2021) contains 21 meteorological features sampled every 10 minutes in Germany, while the Electricity dataset (Wu et al., 2021) tracks hourly electricity consumption of 321 customers. The ILI dataset (Nie et al., 2023) records weekly patient counts and influenza-like illness (ILI) ratios. The Solar dataset (Lai et al., 2018) captures 10-minute intervals of solar power production from 137 PV plants in 2006. Traffic (Wu et al., 2021) dataset records the hourly road occupancy rates from 862 sensors on San Francisco freeways. Additionally, we include the PEMS04 dataset (Liu et al., 2022), which contains public traffic network data from California collected in 5-minute intervals, commonly used in spatio-temporal forecasting. Detailed features and settings of these datasets are presented in Table 4

Table 4: Descriptions of used multivariate time series datasets. Dim column represents the number of variate; and Split column specifies the train-validate-test splitting ratio for each dataset.

| Dataset | Dim | Prediction Length | Split | Frequency | domain |
|---------|-----|-------------------|-------|-----------|--------|
| ETTh1, ETTh2 | 7 | 96, 192, 336, 720 | (6, 2, 2) | Hourly | Electricity |
| ETTm1, ETTm2 | 7 | 96, 192, 336, 720 | (6, 2, 2) | 15min | Electricity |
| Weather | 21 | 96, 192, 336, 720 | (7, 1, 2) | 10min | Environment |
| Electricity | 321 | 96, 192, 336, 720 | (7, 1, 2) | Hourly | Electricity |
| Traffic | 862 | 96, 192, 336, 720 | (7, 1, 2) | Hourly | Transportation |
| Solar | 137 | 96, 192, 336, 720 | (6, 2, 2) | 10min | Energy |
| PEMS04 | 307 | 96, 192, 336, 720 | (6, 2, 2) | 5min | Transportation |
| ILI | 7 | 24, 36, 48, 60 | (7, 1, 2) | Weakly | Health |

# B MODE PRODUCTION

Mode production is a common arithmetic operation in tensor methods (Kernfeld et al., 2015), which relies on two fundamental concepts: tensor folding and tensor unfolding. For simplicity and easy understanding, we provide informal definitions through a 3D tensor example.

Given a real 3D tensor $\mathcal{X} \in \mathbb{R}^{n_1 \times n_2 \times n_3}$, the result of mode-3 unfolding of $\mathcal{X}$ is the matrix $\mathbf{X}_{(3)} \in \mathbb{R}^{n_3 \times n_1 n_2}$, denoted by

$$\text{Fold}_{(3)}(\mathcal{X}) = \mathbf{X}_{(3)}$$

and the mode-3 folding operation recovers the matrix back into the tensor, denoted by

$$\text{Unfold}(\mathbf{X}_{(3)}) = \mathcal{X}$$

We then define the mode-3 production as

$$\mathcal{X} \times_3 \mathbf{M} = \text{Unfold}(\mathbf{M}\mathbf{X}_{(3)}) \in \mathbb{R}^{n_1 \times n_2 \times m}$$

where $\mathbf{M} \in \mathbb{R}^{m \times n_3}$ is the transform matrix. The production can be generalized to any mode of any tensor, leading to the definition of mode-$k$ product.

# C ROBUSTNESS

In order to assess the robustness of our method, we report the standard deviation across four different random seeds. We select four datasets that show marginal improvements in forecasting accuracy compared to the second-best method. The results, presented in Table C, confirm the reliability of the performance outcomes listed in Table 4.1.

Table 5: Robustness of TimeCapsule performance. The results are obtained from four random seeds.

| Dataset | traffic | | weather | | ETTm2 | | electricity | |
|---------|---------|---------|---------|---------|---------|---------|---------|---------|
| Horizon | MSE | MAE | MSE | MAE | MSE | MAE | MSE | MAE |
| 96 | 0.363±0.002 | 0.246±0.001 | 0.143 ±0.001 | 0.189±0.002 | 0.162±0.001 | 0.250±0.000 | 0.126±0.001 | 0.218±0.000 |
| 192 | 0.383±0.000 | 0.257±0.000 | 0.189±0.003 | 0.233±0.002 | 0.218±0.001 | 0.289±0.002 | 0.146±0.001 | 0.238±0.001 |
| 336 | 0.394±0.004 | 0.264±0.006 | 0.241±0.002 | 0.274±0.002 | 0.270±0.001 | 0.324±0.001 | 0.162±0.001 | 0.255±0.001 |
| 720 | 0.430±0.000 | 0.282±0.001 | 0.310±0.002 | 0.326±0.002 | 0.347±0.002 | 0.376±0.002 | 0.195±0.001 | 0.285±0.001 |

# D    MORE STUDIES

## D.1    LOOKBACK WINDOW

This study examines the impact of varying the lookback window on forecasting performance. As illustrated in Fig. 5, the accuracy consistently improves with an enlarged lookback window, ranging from 96 to 512. However, this effect diminishes for small and medium datasets as the window length increases.

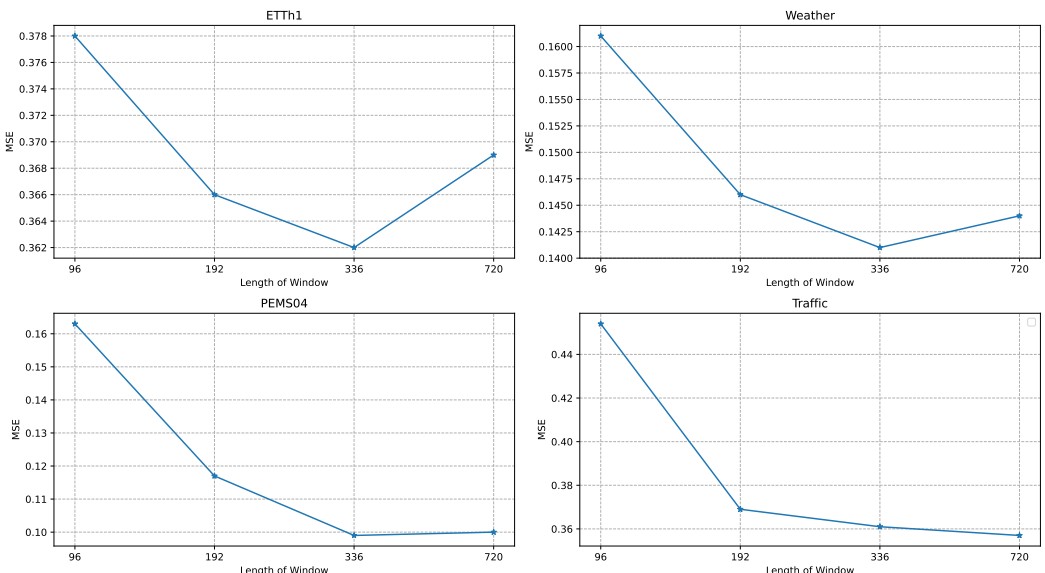

Figure 5: Study on varying lookback windows. We set the length of window $T_x \in \{96, 192, 336, 720\}$, and fix the forecasting horizon $T_y = 96$.

## D.2    NOISE IN ENCODING TUNNELS

To enhance the robustness of the encoding process, we introduced Gaussian noise before the transformation step, as illustrated in Fig. 2 and detailed in formula (12). To evaluate the impact of this added noise, we conducted an ablation study, which is discussed below.

Table 6: Ablations on the noise added in the encoder. The results are obtained by averaging from all four prediction horizons.

| Method | ETTm2 | | Weather | | Traffic | | Electricity | |
|--------|-----------|-----------|-----------|-----------|-----------|-----------|-----------|-----------|
| | MSE (avg) | MAE (avg) | MSE (avg) | MAE (avg) | MSE (avg) | MAE (avg) | MSE (avg) | MAE (avg) |
| Origin | 0.247 | 0.308 | 0.219 | 0.253 | 0.392 | 0.262 | 0.157 | 0.249 |
| w/o noise | 0.250 | 0.310 | 0.224 | 0.258 | 0.392 | 0.262 | 0.219 | 0.265 |

As shown in Table 6, introducing noise generally enhances forecasting performance across the studied datasets. However, the impact varies significantly. For datasets such as ETTm2 and Weather, the improvements are marginal. In contrast, for a large-scale dataset like Traffic, the forecasting results remain unchanged. Notably, for datasets such as Electricity, the improvements are substantial.

### D.3 INHERENT ORDER OF TRANSBLOCKS WITHIN THE ENCODER

We have illustrated the rationale behind the order of TransBlocks in the encoder in Section 3.1. To provide further evidence of its reasonability, we undertake an empirical comparison of the results obtained by TimeCapsule with different order settings. Furthermore, in order to maintain the advantage of efficiency, we only demonstrate the performance variations by exchanging the order of L-Block and V-Block. The results on three large datasets are recorded in the following table, which demonstrates that TimeCapsule with the current order exhibits a slight superiority over TimeCapsule with an exchanged block order. This phenomenon serves to validate our assertions regarding the advantage of learning multi-level properties in advance.

Table 7: Ablations on the block order within the encoder. The results are obtained by averaging from all four prediction horizons.

| Method | Weather | | Traffic | | Electricity | |
|---|---|---|---|---|---|---|
| | MSE (avg) | MAE (avg) | MSE (avg) | MAE (avg) | MSE (avg) | MAE (avg) |
| Origin | 0.219 | 0.253 | 0.392 | 0.262 | 0.157 | 0.249 |
| Exchanged V-L order | 0.223 | 0.261 | 0.407 | 0.274 | 0.165 | 0.260 |

### D.4 POSITIONAL ENCODING

Table 8: Ablations on positional encoding, where w/o PE denotes positional encoding. Performance values are averaged from all four forecasting horizons. The results show that the model's performance decreases when the positional encoding is removed.

| Method | ETTm2 | | Weather | | ETTh2 | | ETTm1 | |
|---|---|---|---|---|---|---|---|---|
| | MSE (avg) | MAE (avg) | MSE (avg) | MAE (avg) | MSE (avg) | MAE (avg) | MSE (avg) | MAE (avg) |
| Origin | 0.247 | 0.308 | 0.219 | 0.253 | 0.338 | 0.386 | 0.345 | 0.376 |
| w/o PE | 0.252 | 0.312 | 0.222 | 0.255 | 0.346 | 0.391 | 0.349 | 0.379 |

In our default settings, temporal positional encoding is applied at the head of the T-TransBlock. We question whether attaching this auxiliary information remains beneficial in the compressed representation space, as its effect could also be an obscure signal to evidence the efficacy of our information compression mechanism. The ablation results, presented in Table 8, show that excluding positional encoding will lead to a slight decline in performance. This suggests that although the impact is minor due to the reduced dimensionality, positional encoding still contributes valuable temporal information even in the compressed representation space.

We can speculate that the effectiveness of positional encoding stems from the linear nature of our compression, which is simply implemented through multiplication by a low-rank transform matrix. This can be illustrated by the following toy example:

$$\mathbf{M}(X + PE) = \mathbf{M}(X) + \mathbf{M}(PE)$$

where $\mathbf{M}$ represents the linear operator, and X and PE correspond to the input data and additive positional encoding, respectively. This demonstrates that the positional encoding is projected into the same compressed transformation space as the input data.

### D.5 COMPRESSION DIMENSION

We provide additional experimental results exploring the Transformer-MLP trade-off by varying the compression dimensions. As shown in Table 9, for most datasets, a robust MLP architecture

Table 9: Full results of study on compression dimensions. Dimension set contains lengths of compression dimension of {T(temporal), V(variate), L(level)}. MSE and MAE are both reported by averaging those from all prediction horizons, and ╱ denotes performance decline while — means no significant changes in performance.

| Dimension Set | origin | | (-, 1, -) | | (1, 1, 1) | | |
|---|---|---|---|---|---|---|---|
| Metric | MSE (avg) | MAE (avg) | MSE (avg) | MAE (avg) | MSE (avg) | MAE (avg) | Trend |
| PEMS04 | 0.120 | 0.223 | 0.136 | 0.236 | 0.156 | 0.257 | ╱ |
| Traffic | 0.392 | 0.262 | 0.404 | 0.277 | 0.410 | 0.285 | ╱ |
| Weather | 0.219 | 0.253 | 0.220 | 0.255 | 0.220 | 0.254 | — |
| Electricity | 0.157 | 0.249 | 0.158 | 0.251 | 0.159 | 0.250 | — |
| Solar | 0.191 | 0.243 | 0.193 | 0.244 | 0.191 | 0.242 | — |
| ETTh1 | 0.408 | 0.428 | 0.413 | 0.430 | 0.419 | 0.433 | ╱ |
| ETTm1 | 0.345 | 0.376 | 0.348 | 0.377 | 0.348 | 0.378 | — |
| ETTh2 | 0.339 | 0.387 | 0.356 | 0.396 | 0.357 | 0.398 | ╱ |
| ETTm2 | 0.248 | 0.309 | 0.251 | 0.311 | 0.250 | 0.311 | — |

without explicit dependency capturing is sufficient to achieve relatively strong performance. This supports our argument that forecasting models often suffer from inefficient resource allocation when processing diverse datasets. A flexible structure, such as that of TimeCapsule, proves effective in adapting to different scenarios.

# E  WHAT HAVE LEARNT BY TIMECAPSULE ?

One of the most exciting aspects of TimeCapsule is its ability to handle time series decomposition autonomously through neural networks, potentially at the expense of interpretability. In this part, we explore the decomposition strategy employed by TimeCapsule through visualizations, aiming to both clarify the inner workings of our model and inspire further investigations into time series modeling with deep learning.

We choose the ETTm2 and PEMS04 datasets to represent, respectively, simpler and more complex time series patterns. Our investigation centers on addressing three key questions:

1. Do the transforms retain and differentiate variable and level information ?

2. What do the transform matrices reveal ?

3. What kind of decomposition strategies that TimeCapsule has learnt ?

## E.1  DO THESE TRANSFORMS RETAIN AND DISTINGUISH THE INFORMATION OF VARIABLES AND LEVELS ?

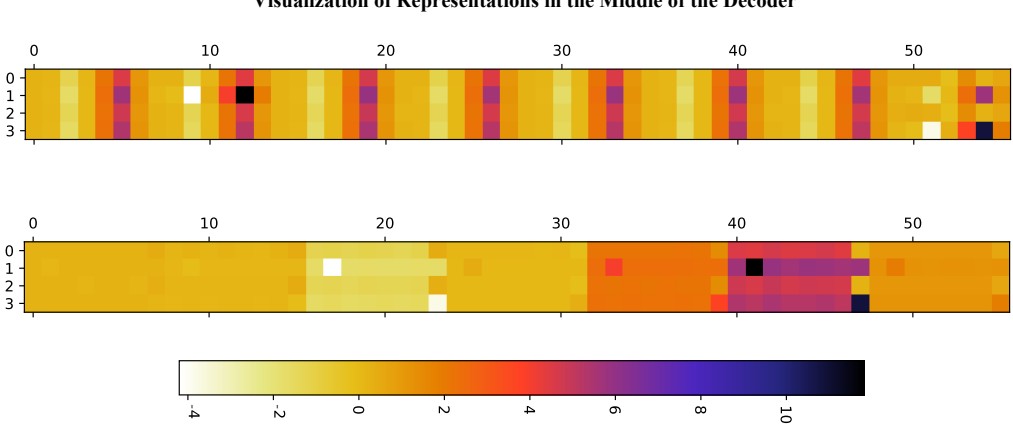

Figure 6: Representaion of ETTm2 in the encoder. We select $\mathcal{X}_2 \in \mathbb{R}^{4 \times 8 \times 7}$ at the end of the L-TransBlock. It has the compressed temporal dimension $t_c = 4$, expanded level dimension $l_c = 8$, and variable dimension $v = 7$.

Figure 7: Representaion of ETTm2 in the decoder. We select $\mathcal{Y}_1 \in \mathbb{R}^{4 \times 8 \times 7}$, which also has the compressed temporal dimension $t_c = 4$, expanded level dimension $l_c = 8$, and variable dimension $v = 7$.

Firstly, as shown in Fig. 6, we present the representation obtained after the time compression and level expansion, denoted as $\mathcal{X}_2 \in \mathbb{R}^{v \times T_c \times l}$ with $v = 4$, $t_c =$,4 and $l = 8$. By folding it into the matrix $\mathbf{X} \in \mathbb{R}^{t_c \times lv}$, we observe that even in the compressed representation space, distinct characteris-

tics of variables remain. These are organized into seven recognized blocks, each may corresponding to a variable in the ETTm2 dataset. What's more, each block contains eight componets, which can be interpreted as level tokens.

In the decoder stage, we examine the representation $\mathcal{Y}_1 \in \mathbb{R}^{v \times T_c \times l}$ in the same way. As illustrated in the bottom half of Fig. 7, though patterns differ from those in $\mathcal{X}_2$, there are still seven blocks. Furthermore, when we swap the dimension of $l$ and $T_c$ in $\mathcal{Y}_1$ (top part of Fig. 7), the representation turns out to have eight blocks with seven components within each, which exactly align with our layout depicted in the leftmost part of Fig. 2. This finding is intriguing: despite the compression of the variable dimension, and without explicitly instructing TimeCapsule to learn level and variable tokens separately, it inherently does so. This suggests that the representation is learned in a predictive and structured manner, indicating that the mode-specific self-attention mechanism is functioning and that multi-level dependencies are effectively captured.

Besides, Fig. 7 also reveals that most levels appear redundant, explaining the results in Table. 9, where multi-level modeling shows minimal benefit for ETTm2.

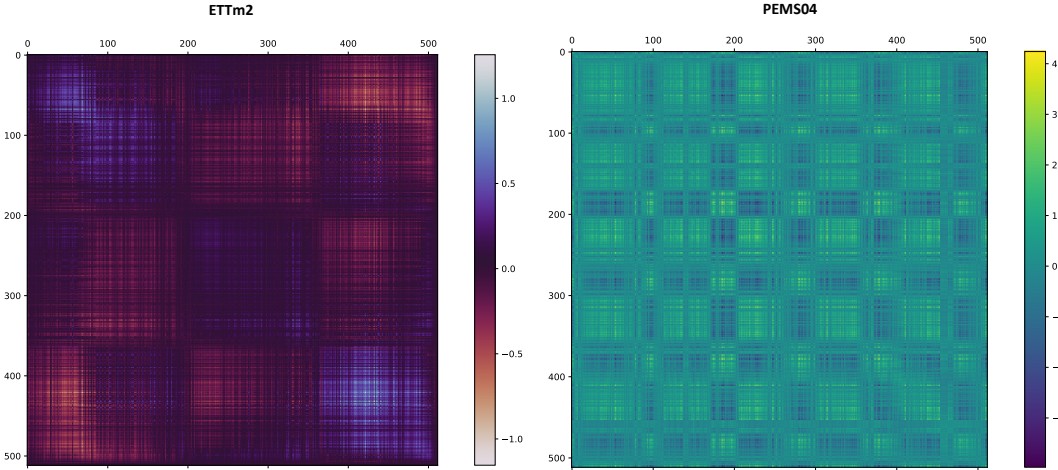

Figure 8: Visualization of T (temporal)-transform matrices, i.e., $\mathbf{M}_T \mathbf{M}_T^\top$. The left part shows the learned temporal pattern of ETTm2 dataset, while the right part shows that of PEMS04 dataset.

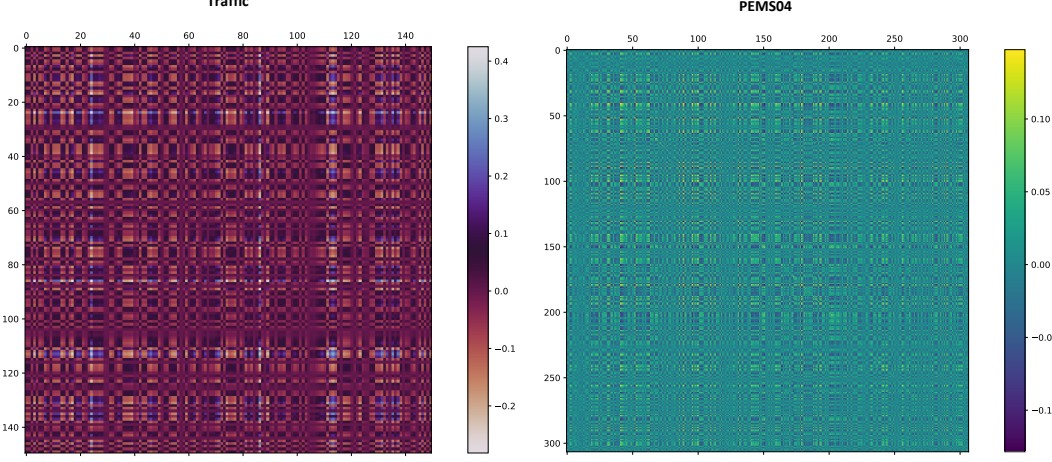

Figure 9: Visualization of V (variable)-transform matrices, i.e., $\mathbf{M}_V \mathbf{M}_V^\top$. Since the number of variables contained in ETTm2 are too small to demonstrate a significant pattern, we replace it with Traffic dataset.

## E.2 What do these transform matrices look like to enable multi-mode and multi-level learning ?

The capacity for information compression reflects the model's ability to capture and aggregate dependencies. As expected, these matrices should exhibit a certain block structure to capture independent and hierarchical features. For clarity, we denote the transform matrices as $\mathbf{M}$ and present them in symmetric form as $\mathbf{M}\mathbf{M}^\top$. We omit the level expansion transformation matrix, as its $\mathbf{M}\mathbf{M}^\top$ results in a scalar value.

As illustrated clearly in Fig. 8 and Fig. 9, these low-rank transforms exhibit significant patterns, demonstrating our model's strategies to capturing temporal and variable dependencies within time series. These visualizations of learned compression/expansion matrices hold promising potential for analyzing the unique characteristics of different time series.

## E.3 What decomposition strategies has TimeCapsule learned ?

We have observed that TimeCapsule recognizes the multi-level structure of time series, although this property resides in the latent representation space. In order to gain insight into it, we visualize the final representation $\mathcal{Y}_3 \in \mathbb{R}^{v \times t_y \times 1}$ at the neck of the decoder. By applying the learned level expansion transform denoted as $\mathbf{M}_L \in \mathbb{R}^{1 \times l}$, we decompose $\mathcal{Y}_3$ into $l$ sub-level series by performing the mode-3 product $\mathcal{Y}_3 \times_3 \mathbf{M}_L$, then display each one, fixing on the first variable. As seen in Fig. 11 and Fig. 10, these series generally exhibit different scales, indicating that the multi-level property contains a multi-scale property within the representation space. Specifically, for the ETTm2 dataset (see Fig. 11), each level's series has a unique amplitude, and their frequencies group into different ranges; whereas for PEMS04 (see Fig. 10), the level patterns appear clearer and simpler.

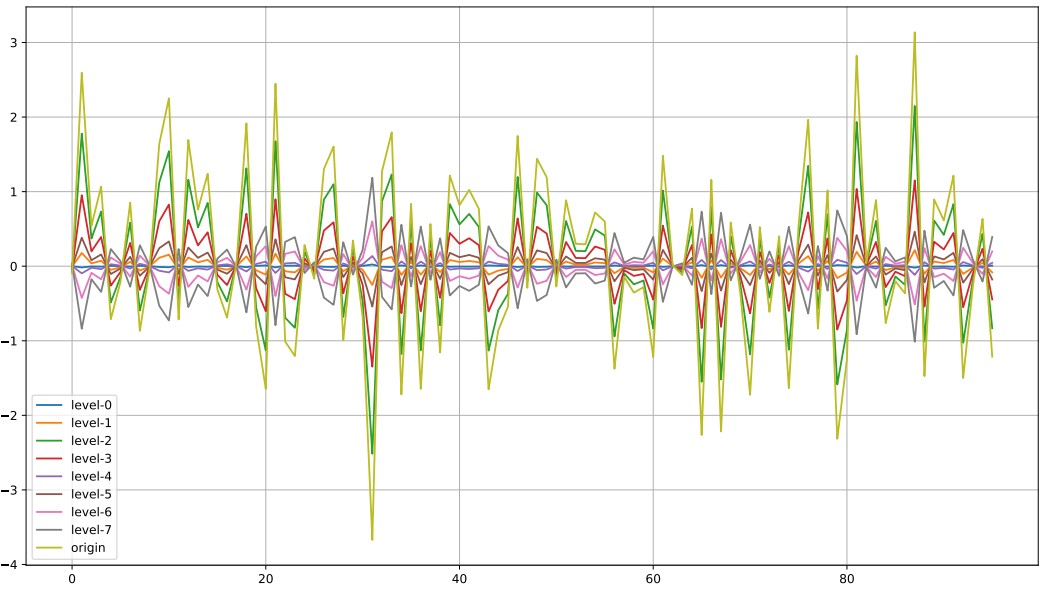

Figure 10: Multil-level property of PEMS04 series in the representation space by applying the learnd level-expansion matrix.

This observation raises another question: is this multi-scale effect a result of the level expansion transform $\mathbf{M}_L$ ? To figure out it, we apply the transform again to $\mathcal{X}_0$, the original time series in the time domain, just after the instance normalization. We decompose it into the same number of sub-level series. As shown in Fig. 12 and Fig. 13, these decomposed sub-series also exhibit different scales and frequencies. This demonstrates the effectiveness of the learned level expansion transform and suggests that we could leverage such transforms to generate new kinds of time series decompositions.

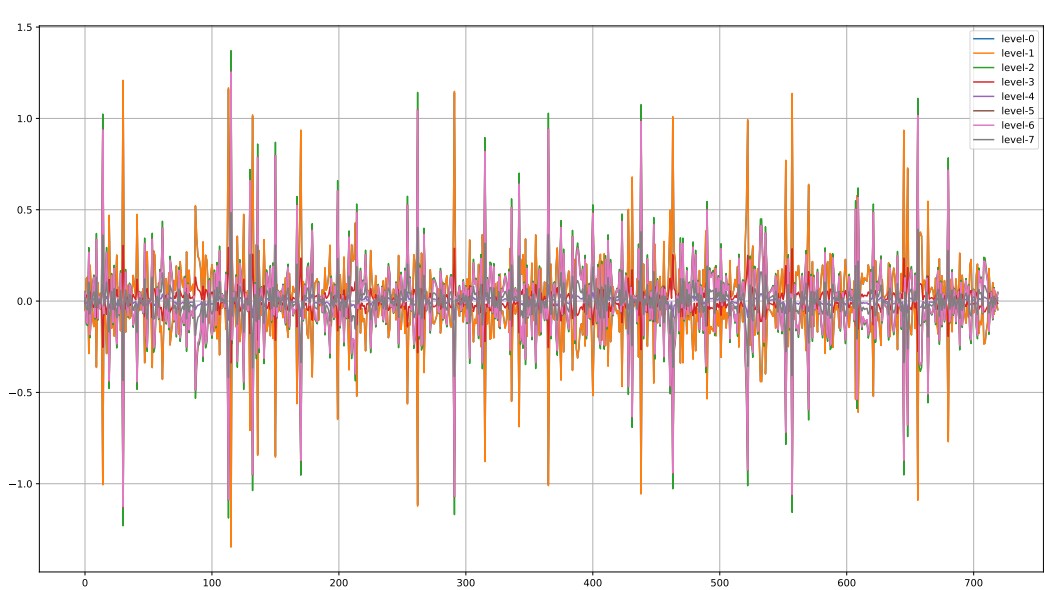

Figure 11: Multil-level property of ETTm2 series in the representation space by applying the learnd level-expansion matrix.

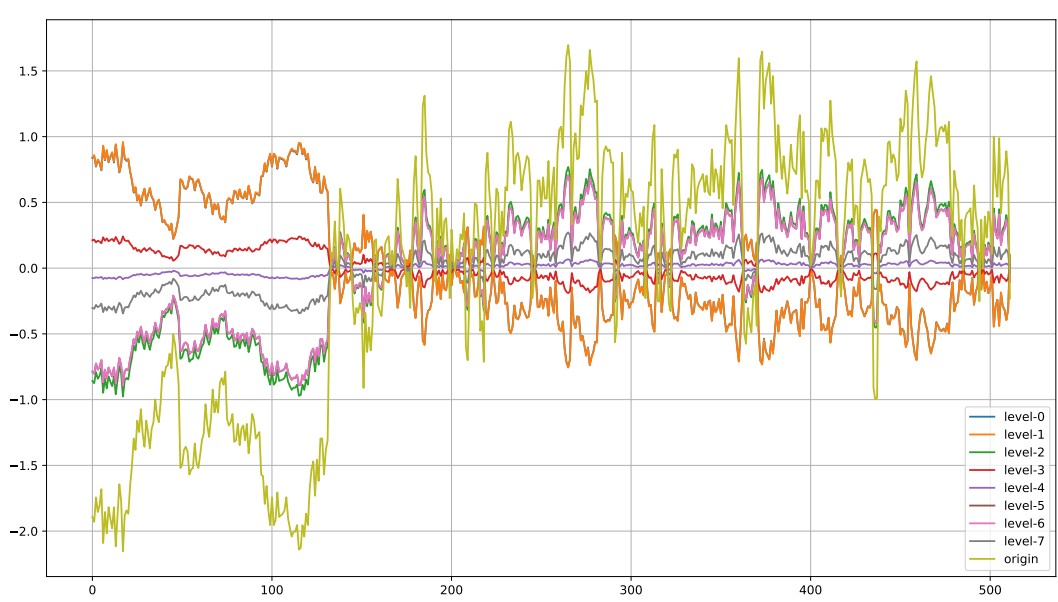

Figure 12: Multil-level property of ETTm2 series in the normalized time space by applying the learnd level-expansion matrix.

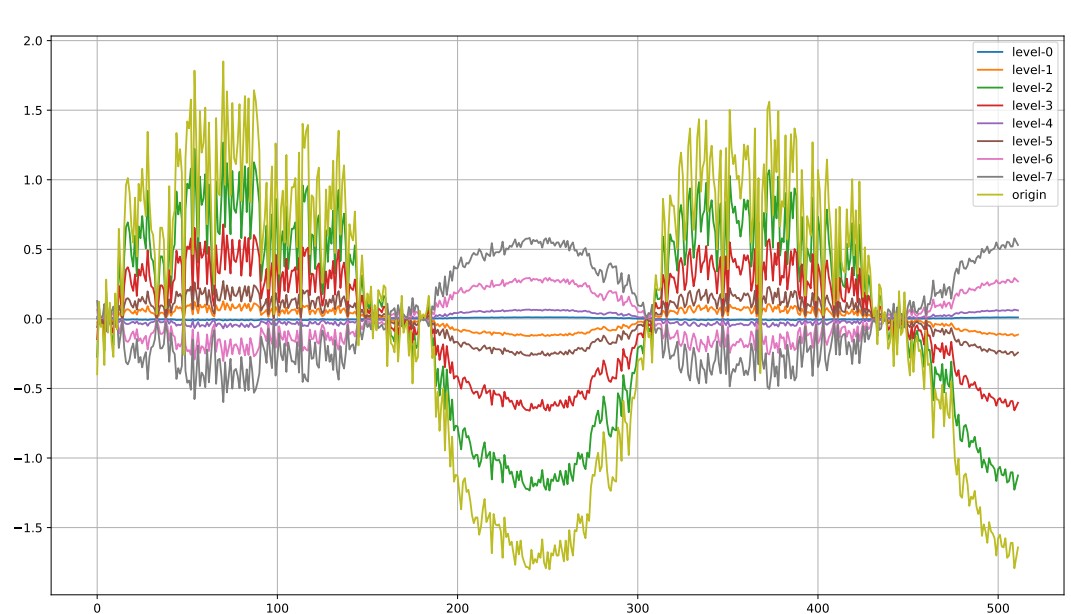

Figure 13: Multil-level property of PEMS04 series in the normalized time space by applying the learnd level-expansion matrix.

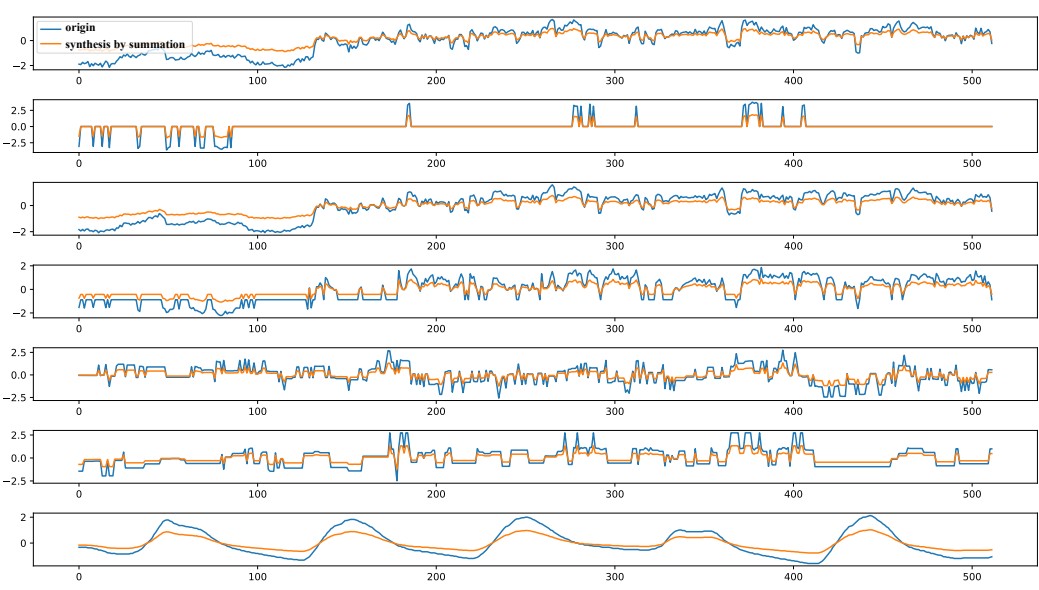

Figure 14: A summation of the sub-level series of the ETTm2 series in the normalized time space demonstrates that it approaches the original series, indicating that the linear decomposition has been achieved by the learned level-expansion matrix. Each subfigure represents a distinct variable of the ETTm2.

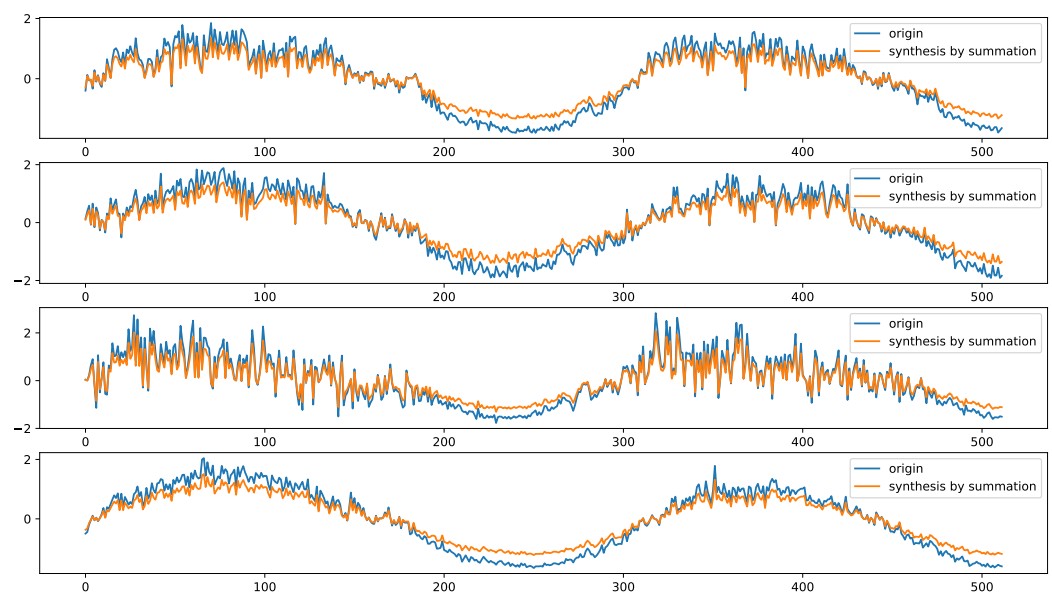

Figure 15: A summation of the sub-level series of the ETTm2 series in the normalized time space demonstrates that it approaches the original series, indicating that the linear decomposition has been achieved by the learned level-expansion matrix. The first four variables are demonstrated in the figure.

Finally, although we refer to the level expansion as a kind of time series decomposition, we need to check that these sub-level series are indeed the result of a certain decomposition. As a naive endeavor, we sum all the sub-level series of the decomposed $\mathcal{X}_0$ into a single series. Suprisingly, the summation result nearly reconstructs the original series, as shown in Fig. 14 and Fig. 15. This validates that our model's $\mathbf{M}_L$ achieves a linear decomposition for time series. Moreover, it also suggests that these levels act as time bases, similar to the basis functions discussed in models like N-BEATs (Oreshkin et al., 2019) and N-Hits (Challu et al., 2023). Broadly speaking, each sub-level captures a unique aspect of the series: high-frequency sub-series contribute to the coarse structure, while low-frequency sub-series refine finer details. This may also help to explain why linear dependencies play a crucial role in time series modeling.

## F  LIMITATIONS AND FUTURE WORK

As discussed by current LTSF benchmark works (Qiu et al., 2024), no existing model can emerge as the best across all cases. While we are are pleased to find that TimeCapsule can consistently achieve SOTA performance on diverse datasets with competitive computation speeds, its flexibility also introduces many challenges such as hyperparameter tuning. In the following, we elaborate on several limitations of our proposed model, which may stimulates further explorations and refinements.

**Compression Setting:** The model currently relies on fixed, hard-coded compression dimensions as hyperparameters, meaning that it is enforced to compress information into a predefined representation size. This constraint might restrict the model's full potential, especially for datasets that benefit from adaptive compression. Hence, exploring more mechanisms about time series decomposition and the compressibility of time series could lead to more flexible and effective architectures.

**Component Utilization:** TimeCapusle's modular structure sometimes reveals uneven utilization of its components. For instance, transformer blocks for compressed representation learning and dependency capture are highly effective in some cases but may go underused in others. Likewise, the dimension of linear projection within the MLP usually can significantly impact results. This variability points to a potential waste of computational resources when dealing with different datasets. Regarding to this issue, we will aim to analyze each component's role in the model and then distill the structure into a more compact form for practical applications.

**Generality:** The current version of TimeCapsule is specifically tailored to address one of the most important challenges in time series research—long-term time series forecasting (LTSF). Despite the shown initial classification results (see Appendix G.2), its design and functionality are primarily centered around optimizing LTSF, which may inherently limit its expressiveness and effectiveness for other downstream tasks, such as classification and imputation. Exploring TimeCapsule's adaptability to these broader applications, including investigating structural adjustments to better accommodate diverse practical applications, represents an urgent modification direction for the proposed TimeCapsule.

# G A PRELIMINARY INVESTIGATION INTO THE POTENTIAL FOR FURTHER APPLICATIONS

## G.1 TIMECAPSULE PRE-TRAINING USING JEPA

Due to the introduction of JEPA and the explicit division of functionalities within TimeCapsule, where the encoder learns the predictive representation and the decoder makes predictions based on the compressed representation, it is natural to explore its forecasting ability under a pre-training scheme.

Instead of using traditional masked pre-training, we design and test a specialized efficient JEPA-based pre-training and fine-tuning strategy for TimeCapsule. Specifically, in the pre-training phase, we decouple the training process of the encoder and decoder: we first train the encoder using the JEPA loss, and then, with the encoder and representation predictor frozen, train the decoder using only the final prediction loss. Afterward, we fine-tune the entire model by optimizing the sum of the two losses.

We evaluate the performance of this training strategy on several large datasets. We set the maximum pre-training epochs as 10 and 50 for the encoder and decoder, respectively, and fine-tune the entire model for a maximum of 10 epochs. As shown in the following table, the results indicate that this pre-training strategy leads to comparable performance, or even worse results, than supervised learning.

Table 10: Comparison results of TimeCapsule under the mode of JEPA based pre-traning and fine-tuning.

| Models | | TimeCapsule (fine-tune) | | TimeCapsule (supervised) | | iTransformer (2023) | | TimeMixer (2024) | | PatchTST (self-supervised 2023) | | Crossformer (2023) | |
|---|---|---|---|---|---|---|---|---|---|---|---|---|---|
| Metric | | MSE | MAE | MSE | MAE | MSE | MAE | MSE | MAE | MSE | MAE | MSE | MAE |
| Weather | 96 | 0.142 | 0.187 | **0.141** | **0.186** | 0.159 | 0.208 | 0.147 | 0.198 | 0.148 | 0.196 | 0.146 | 0.212 |
| | 192 | 0.188 | 0.233 | **0.187** | **0.232** | 0.200 | 0.248 | 0.192 | 0.243 | 0.193 | 0.240 | 0.195 | 0.261 |
| | 336 | **0.236** | **0.271** | 0.239 | 0.272 | 0.253 | 0.289 | 0.247 | 0.284 | 0.244 | 0.279 | 0.268 | 0.325 |
| | 720 | 0.310 | 0.325 | **0.309** | **0.323** | 0.321 | 0.338 | 0.318 | 0.330 | 0.321 | 0.334 | 0.330 | 0.380 |
| Traffic | 96 | 0.369 | 0.252 | **0.361** | **0.246** | 0.363 | 0.265 | 0.466 | 0.294 | 0.382 | 0.262 | 0.514 | 0.282 |
| | 192 | 0.388 | 0.260 | **0.383** | **0.257** | 0.385 | 0.273 | 0.508 | 0.299 | 0.385 | 0.261 | 0.501 | 0.273 |
| | 336 | 0.396 | 0.265 | **0.393** | **0.262** | 0.396 | 0.277 | 0.526 | 0.309 | 0.409 | 0.275 | 0.507 | 0.278 |
| | 720 | 0.435 | 0.288 | **0.430** | **0.282** | 0.445 | 0.312 | 0.554 | 0.322 | 0.438 | 0.291 | 0.571 | 0.301 |
| Electricity | 96 | 0.127 | 0.221 | **0.125** | **0.218** | 0.138 | 0.237 | 0.131 | 0.224 | 0.132 | 0.227 | 0.135 | 0.237 |
| | 192 | 0.146 | 0.239 | 0.146 | **0.238** | 0.157 | 0.256 | 0.151 | 242 | 0.148 | 0.241 | 0.160 | 0.262 |
| | 336 | 0.163 | 0.256 | **0.161** | **0.254** | 0.167 | 0.264 | 0.169 | 0.260 | 0.167 | 0.260 | 0.182 | 0.282 |
| | 720 | 0.198 | 0.287 | 0.194 | **0.285** | 0.194 | 0.286 | 0.227 | 0.312 | 0.205 | 0.292 | 0.246 | 0.337 |

We attribute this unsatisfactory outcome to two main factors. First, in our default training setup, JEPA is only involved in the training of the encoder. While the prediction loss contributes to parameter updates across the entire model, the overall training effect does not significantly differ from the separated training process. Second, the representation involved in the JEPA loss is compressed into a very small size, which limits its impact on parameter updates and makes it less effective for pre-training. To address this issue, we propose that increasing the influence of JEPA by incorporating additional losses computed between representations at different hidden layers could improve performance, as demonstrated in previous works (e.g., (Baevski et al., 2022)).

## G.2 CLASSIFICATION

In this section we take a first look at the potential of TimeCapsule in an alternative key time series application, classification, which has played a crucial role in many real world scenarios. In order to

achieve this purpose with a minimal change to the model structure, we turn off the use of JEPA due to its unclear interpretation in classification.

Table 11: Results for classificatino task. The classification accuracy ($\%$) are recorded as the results below.

| Methods / Datasets | Informer (2021) | Pyraformer (2021) | Autoformer (2021) | FEDformer (2022b) | iTransformer (2023) | Dlinear (2023) | TiDE (2023) | Timesnet (2022) | TimeCapsule |
|---|---|---|---|---|---|---|---|---|---|
| Heartbeat | 80.5 | 75.6 | 74.6 | 73.7 | 75.6 | 75.1 | 74.6 | 78.0 | 78.5 |
| FaceDetection | 67.0 | 65.7 | 68.4 | 66.0 | 66.3 | 68.0 | 65.3 | 68.6 | 70.2 |
| Handwriting | 32.8 | 29.4 | 36.7 | 28.0 | 24.2 | 27.0 | 23.2 | 32.1 | 27.0 |
| SelfRegulationSCP2 | 53.3 | 53.3 | 50.6 | 54.4 | 54.4 | 50.5 | 53.4 | 57.2 | 57.8 |
| EthanolConcentration | 31.6 | 30.8 | 31.6 | 28.1 | 28.1 | 32.6 | 27.1 | 35.7 | 32.0 |
| UWaveGestureLibrary | 85.6 | 83.4 | 85.9 | 85.3 | 85.9 | 82.1 | 84.9 | 85.3 | 88.8 |
| Average Accuracy | 58.5 | 56.4 | 58.0 | 55.9 | 55.8 | 55.9 | 54.8 | 59.5 | 59.1 |

We select six of the most challenging datasets used in (Wu et al., 2022). The results in Table. G.2 show that although all the modules designed in this paper are primarily dedicated to improving the generality and performance of the model in LTSF, it can still achieve competitive classification accuracies compared to other time series models in recent years, which further underscores the capability of TimeCapsule.

# H  VISUALIZATIONS OF FORECASTING

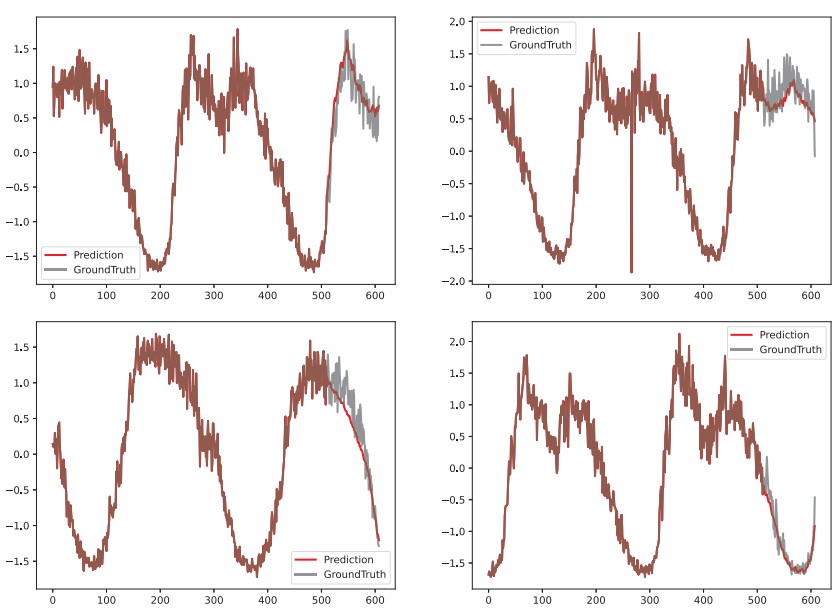

Figure 16: Prediction on the PEMS04 dataset, with lookback window 512 and forecast length 96.

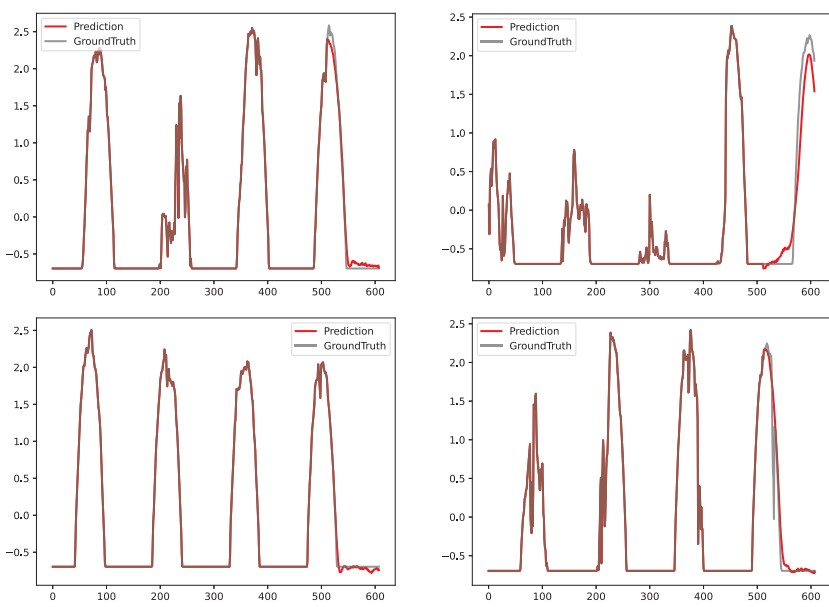

Figure 17: Prediction on the Solar dataset, with lookback window 512 and forecast length 96.

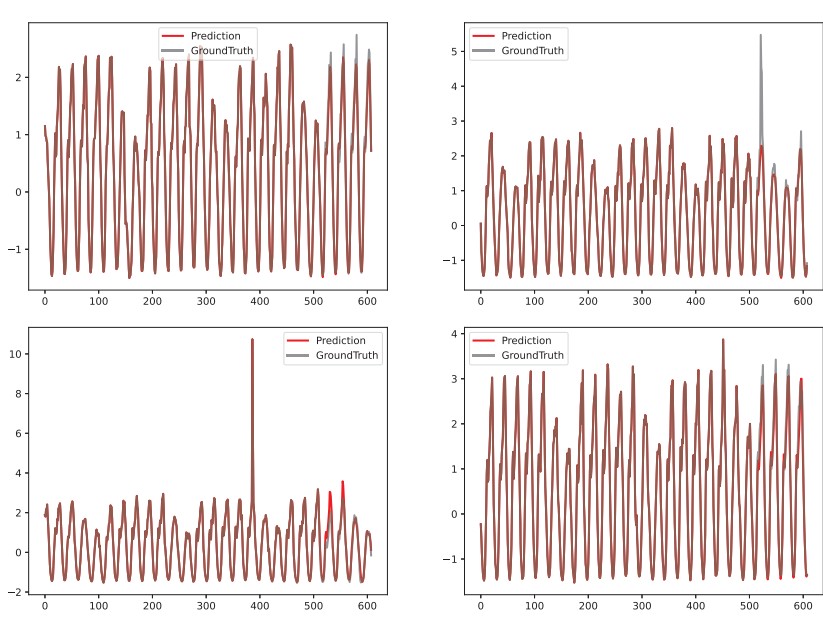

Figure 18: Prediction on the Traffic dataset, with lookback window 512 and forecast length 96.

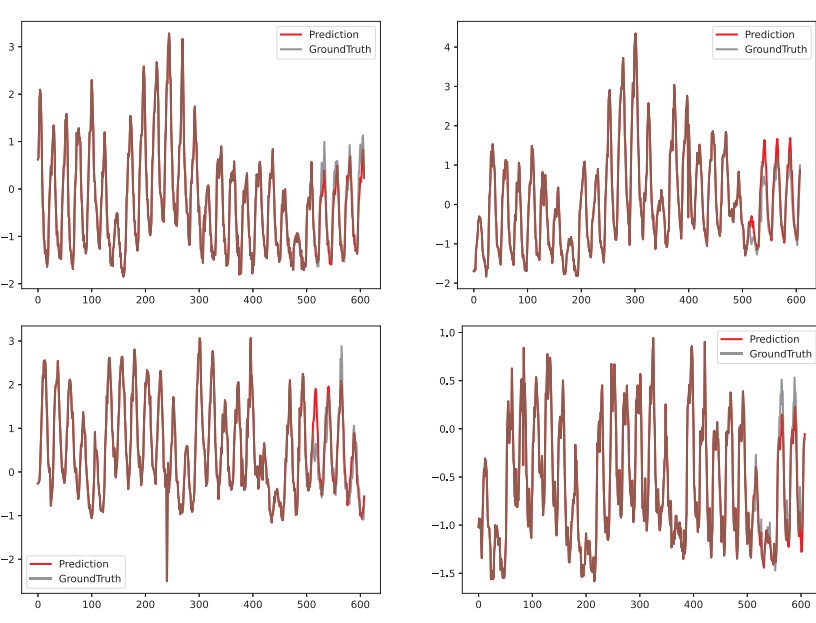

Figure 19: Prediction on the Electricity dataset, with lookback window 512 and forecast length 96.

