# OpenReview forum: "TimeCapsule:  Solving the Jigsaw Puzzle of Long-Term Time Series Forecasting with Compressed Predictive Representations"
_ICLR.cc/2025/Conference — ICLR 2025 Conference Withdrawn Submission_

### Official Review · Reviewer_Nedg · 2024-10-29

**Soundness:** 3
**Presentation:** 3
**Contribution:** 2
**Rating:** 5
**Confidence:** 5

**Summary:**

This paper integrates existing time series analysis methods including temporal, variate, multi-scale, etc. to deal with various multivariate time series scenarios. It proposes a corresponding preprocessing and prediction method framework. And applies the JEPA method to the highly compressed representation space and adds the corresponding loss.

**Strengths:**

The research direction of the article is relatively innovative. Indeed, integrating various current methods is a good direction, and the methods used are logically reasonable. The article's expression and illustrations are overall very clear.

**Weaknesses:**

1. I think there is a certain difference between the Target and context inputs of JEPA in this article and, for example, I-JEPA. In I-JEPA, the context and Target overlap. Here, I think it is worthy of further demonstration. Based on more sufficient experimental evidence, a more appropriate design should be given. I think the application of JEPA to TS problems here is not studied in-depth.
2. I think it is necessary to compare the effects and efficiencies of MLP and the Transformer Decoder. The lack of experiments means that it cannot be directly concluded as unreasonable.
3. For the order of TransBlocks and the comparison with the parallel method, there is a lack of experimental support.
4. The elaboration of Decomposition classification and level is not very clear. I think adding some visualizations or results after the processing of each TransBlock (T,V,L) and the analysis of how to correspond to the function will be more clear and intuitive.
5. I think the method proposed in the article has some limitations. For example, the selection of the lookback window is an unconventional 512. Of course, the article also analyzes different lengths. But it is obvious that the larger is more effective. I suggest clarifying this point and whether there are other limitations. I think limitations are also a contribution to the development of the field.
6. No code is provided for reproducibility.

**Questions:**

1. Line 281-283 Utilizing JEPA, what specific inductive bias is introduced?
2. Why is TransBlocks in this order? Is there a difference in different processing orders for different types of datasets? What is the difference compared to parallel processing?
3. Why only select one subset, PEMS04 of PEMS? I think the advantages and limitations of a method should be supported by comprehensive experimental results, rather than conducting a large amount of analysis based on the performance on a single example dataset as in the Main Results part.

---

> ### Author Response · Authors · 2024-11-21
>
> Dear Reviewer Nedg,
>
> We sincerely appreciate your identifying our work's novelty and providing thorough detailed comments and suggestions. Here are our answers to your concerns and questions:
>
> **Q1 & W1: About JEPA used in our framework**
>
> Thank you for this valuable question. We’re glad to provide further insights into our rationale for using JEPA in this framework.
>
> First, we want to clarify here why we chose to incorporate JEPA. JEPA (Joint-Embedding Predictive Architecture) was originally proposed as a natural approach for learning predictive representations [1]. In our framework, the weak transformers serve as the encoder to process time series information and capture multimodal dependencies. This results in a compressed representation of the **history information** (which you referred to as the context). The MLP decoder then utilizes this compressed information to model the time series and generate **future predictions**.
>
> **The input sequence actually shares the same data space as the output sequence since they are both from the same time series.**  However, there may be a gap between the historical information and the future forecasts (as discussed in lines 294~297 in the revised version), making it difficult to directly use the historical information for prediction. This distinction is one of the key features that sets TimeCapsule apart from previous forecasting frameworks. To address this gap, we introduce a **representation prediction** step, where we use JEPA loss to align the history and future representations. This predictive alignment justifies the use of JEPA in our model.
>
> From a broader perspective, we view this additional representation learning in the forecasting process as an augmentation technique for LTSF. The effectiveness of this augmentation depends on factors such as the information content of the original time series and the overall architecture of the model. We discussed the impact of JEPA in the **Effects of JEPA and Predictive Representation** section of the experiments, where we carefully tracked and compared the variations in JEPA loss, both with and without backpropagation. We also provided the inductive biases introduced by this loss function, according to our understanding, from an optimization perspective. Shortly, the JEPA loss function helps bridge the gap between the historical information landscape and the future prediction landscape. Minimizing this distance during training typically improves performance, although its significance can vary depending on the specific dataset and context.
>
> We sincerely appreciate the reviewers' attention to our use of JEPA. While it is an important component in our framework, it is not the central focus of this research. We plan to explore its effects and potential applications further in future work.
> ****
> **W2:  About the effects and efficiencies of MLP and the Transformer Decoder**
>
> Thank you for your insightful question. First, we would like to clarify that the asymmetric structure in TimeCapsule is a deliberate design choice aimed at integrating and exploring the effect of the **generalized linear projection** technique (one of the groups in our categorization). If we were to replace the MLP decoder with another architecture, the technique we intend to include in our model would lose its relevance.
>
> While it is true that we cannot definitively conclude that the MLP decoder is better than a transformer-based one, such a substitution would go against the primary motivation of our research. Our goal is to investigate a trade-off between transformer and linear modules. As explained in the **Introduction** (lines 84~92), transformers are employed in our model as the **predictive representation learners**, while the MLP decoder is used to model the **linear dependencies** within the time series and generate the forecasting results.
>
> In essence, this architecture is designed to explore the complementary roles of the transformer encoder and the linear decoder. This investigation is discussed in detail in the **Analysis of Asymmetric Structure** section of the manuscript. Hence, this is why we did not consider swapping the MLP decoder for a transformer-based one in our experiments.
> ****
> **W3: About the sequence structure and order within the encoder**
>
> The explanations about this design are listed in our answer to **Q2**.
>
> [1] LeCun, Yann. "A path towards autonomous machine intelligence version 0.9. 2, 2022-06-27." Open Review 62.1 (2022): 1-62.

---

> ### Author Response · Authors · 2024-11-21
>
> **W4:" Some visualizations or results after the processing of each TransBlock (T, V, L) and the analysis of how to correspond to the function will be more clear and intuitive"**
>
> Thanks for this extremely valuable suggestion! In response, we have added a specific and detailed visualization-based exploration in **Appendix E**, where we answer three key questions. We hope that these new findings will be helpful in improving the quality and clarity of our research.
> ****
> **W5:I think the method proposed in the article has some limitations. For example, the selection of the lookback window is an unconventional 512. Of course, the article also analyzes different lengths. But it is obvious that the larger is more effective. I suggest clarifying this point and whether there are other limitations. I think limitations are also a contribution to the development of the field.**
>
> Thanks for your thoughtful question and valuable suggestion. We completely agree with your point that “limitations are also a contribution to the development of the field.”, so we have included a limitation analysis in **Appendix F**.
>
> Regarding the selection of the lookback window size of 512:
>
> - **Common Choice in LTSF Research:** The length 512 is commonly used in many LTSF papers, such as PatchTST, and has been shown to work well in various benchmarks.
> - **Effectiveness of Larger Windows:** We respectfully disagree with the reviewer’s opinion that a larger window size is always more effective. For many LTSF models, longer lookback windows can be problematic, as they may introduce more noise and redundancy into the data, and also may cause overfitting. Additionally, capturing long-term dependencies effectively becomes more challenging as the window size increases. In fact, the ability to leverage long historical information is an important indicator of a model’s effectiveness, and our results with the 512 window demonstrate TimeCapsule’s capacity in this regard.
> - **Benchmark Consistency:** The comparison benchmark used in our study, TFB, is a very challenging LTSF benchmark whose results were obtained by thoroughly searching various lookback windows in the range of [96, 336, 512, 720]. Thus, to ensure a fair comparison across models, we fixed the window size at 512, which was necessary for consistency, rather than a design choice.
> ****
> **W6: About the source code**
>
> Thank you for your attention to this matter. We chose not to publish our code before to avoid the risk of potential information leakage that could lead to desk rejection. However, we assure you that we will release the code shortly after the conference, ensuring the full reproducibility of our results. We are working on organizing the messy parts of the source code. If you have any considerations about the empirical results, we will gladly share the code as soon as possible.

---

> ### Author Response · Authors · 2024-11-21
>
> **Q2:Why is TransBlocks in this order? Is there a difference in different processing orders for different types of datasets? What is the difference compared to parallel processing?**
> The order of transformations—T (time), L (level), and V (variate)—is a result of careful design rather than arbitrary choice. Specifically:
>
> - **T (Time)** comes first because the input length of the time series is typically very large. If this dimension were handled later in the process, it could lead to prohibitive computational costs during multimodal self-attention computation. By applying dimensionality compression to the time dimension first, we can mitigate this issue.
> - **L (Level)** is handled second, as it is essential for multi-level learning in the representation space. It would be too late to expand the level dimension at a later stage, so we place this step after time compression.
> - **V (Variate)** is the last transformation because it involves fine-tuning the representation after the time and level dimensions have been processed. This sequence ensures that each dimension is processed in a logical order that supports the overall model architecture.
>
> To improve the readability and help understanding of this design, **we add some descriptions and intuitive explanations in lines 213 ~ 215.(of revised version)**
>
> Regarding the question of parallel processing, please refer to our response to **Q2 of Reviewer MW4b**, where we answered the same concern.
> ****
> **Q3: Why only select one subset, PEMS04 of PEMS? I think the advantages and limitations of a method should be supported by comprehensive experimental results, rather than conducting a large amount of analysis based on the performance on a single example dataset as in the Main Results part.**
>
> Thank you for your thoughtful concerns. As a new method for LTSF, the primary advantage of TimeCapsule is evident in the extensive table of comparison results, where our model consistently achieves or approaches the best performance across all datasets and forecasting horizons. This is strong evidence of TimeCapsule's forecasting ability, especially considering the extremely challenging benchmark (TFB [1]) we selected. The detailed results can be found in the **Main Results** section. Further advantages of the model are discussed in the **Model Analysis** section.
>
> We also present a specific case study in the benchmark, using **PEMS04** to demonstrate the effectiveness of multi-scale modeling for spatiotemporal datasets (lines 380–385 in the revised version). This allows us to evaluate TimeCapsule’s performance and compare it with other models. However, aside from this, our conclusions are primarily derived from the comparison of models across all datasets, and we do not treat PEMS04 as a special focus for analyzing other model components. While PEMS04 is included in several studies due to its significant improvement with TimeCapsule, we do not single it out for broader conclusions. For example, we use **ETTh2**, **ETTm2**, **PEMS04**, etc., to explore the effects of residual information compensation, and datasets like **ETTm2**, **weather**, **electricity**, and **PEMS04** to analyze the effects of JEPA.
>
> As for the specific choice of **PEMS04** from the PEMS dataset, this is due to the benchmark (TFB) we selected, which conducted experiments only on **PEMS04** and **PEMS08** from PEMS. While we obtained satisfying results on **PEMS08**, some of the results from this dataset in the benchmark could not be replicated. Furthermore, the discrepancy between the reported results and those we obtained using the open-sourced code is notably large. To avoid any concerns about the experimental results, we decided not to include **PEMS08** in our final report, given that the number of datasets included in our experiment is already sufficiently diverse.
>
> [1] Xiangfei Qiu, Jilin Hu, Lekui Zhou, Xingjian Wu, Junyang Du, Buang Zhang, Chenjuan Guo, Aoy-
> ing Zhou, Christian S Jensen, Zhenli Sheng, et al. Tfb: Towards comprehensive and fair bench-
> marking of time series forecasting methods. arXiv preprint arXiv:2403.20150, 2024.
>
> Authors

---

> > ### Comment · Reviewer_Nedg · 2024-11-25
> >
> > If the reasons for the order（Q2） here are sufficient, they should be explained in the text. However, ablation is more important evidence than insight.

---

> ### Author Response · Authors · 2024-11-25
> **Rebuttal Deadline Approaching**
>
> Dear Reviewer Nedg,
>
> We've answered all your questions and made modifications based on your suggestions. We sincerely appreciate your thoughtful questions and suggestions, which have taken our manuscript to the next level. We hope we've addressed all your concerns, and we'd love to hear your feedback. Please let us know if you have any further questions.
>
> Yours sincerely,
>
> The Authors

---

> ### Author Response · Authors · 2024-11-25
> **Thanks for your reply !**
>
> We would like to express our gratitude for your reply and further inquiries. As previously mentioned, the rationale behind the ordering of the transform blocks can be found in Section 3, specifically in lines 213 to 215.
>
> We agree with your opinion that ablation may be more convincing. However, the order of the sequence is rather than an insight. For example, if we place the T block later, the computational cost will undoubtedly be much more expensive, reducing the efficiency advantage of our model. Concerning the order of L and V, to be honest, there is no obvious distinct based on our experimental observation, the performance was compared initially when we constructed the model, which evidenced that by placing L before, TimeCapusle performs slightly better. We attribute this to the model's ability to rapidly learn multi-level properties,  as we have investigated in Appendix E according to your suggestion in weakness 4. We believe that our additional visualizations and analyses can provide a more comprehensive and intuitive explanation.
>
> Looking forward to your reply.

---

> ### Author Response · Authors · 2024-11-25
> **Reply to the further concern**
>
> Dear Reviewer Nedg,
>
> Although we believe that the order within the encoder is not very important in evaluating the contributions of this research, we'd be happy to run the previous test version of TimeCapsule, where the order of the transform blocks of L and V is transposed. We evaluated this on the following three datasets with a relatively large number of variables. The results are shown in the table below, which shows that TimeCapsule with the current order performs better than TimeCapsule with the exchanged order.
>
> | **Method** |**Weather** (MSE avg) | **Weather** (MAE avg) | **Traffic** (MSE avg) | **Traffic** (MAE avg) | **Electricity** (MSE avg) | **Electricity** (MAE avg) |
> |------------|---------------------|---------------------|-----------------------|-----------------------|-----------------------|-----------------------|
> | **Origin** | 0.219                 | 0.253                 | 0.392                 | 0.262                 | 0.157                     | 0.249                     |
> | **Exchanged L-V order** | 0.223                 | 0.261                 | 0.407                 | 0.274                 | 0.165                     | 0.260                     |

---

> ### Author Response · Authors · 2024-11-27
>
> Dear Reviewer Nedg,
>
> Thanks again for your valuable and constructive review. Following your suggestions and concerns, we have added additional empirical results and many visual analyses that significantly improve the overall quality of the manuscript. However, we note that the overall score has dropped to the rejection threshold. We kindly hope you can consider our response and the efforts we have made for the modifications. If there are any further comments from your side, we will be happy to address them before the rebuttal deadline. We would appreciate it if you could reconsider your rating.
>
> Thank you once again for your support and feedback.
>
> Wishing you all the best,
>
> The Authors

---

> ### Author Response · Authors · 2024-12-01
>
> Dear Reviewer Nedg,
>
> As the extended rebuttal period is coming to an end, we would like to ask if it would be possible to consider increasing your score based on the improvements made to the article as a result of your suggestions.
>
> Yours sincerely,
>
> The Authors

---

### Official Review · Reviewer_5FaK · 2024-10-30

**Soundness:** 2
**Presentation:** 2
**Contribution:** 2
**Rating:** 5
**Confidence:** 4

**Summary:**

This paper introduces **TimeCapsule**, a model aimed at simplifying and unifying key techniques in long-term time series forecasting (LTSF) while achieving comparable performance to complex, state-of-the-art models. TimeCapsule models time series as a 3D tensor, incorporating temporal, variate, and level dimensions, and leverages mode production to capture multi-mode dependencies while achieving dimensionality compression. Experiments on challenging benchmarks highlight TimeCapsule's effectiveness.

**Strengths:**

1. This paper provides a new insight into treating time series as a 3D tensor.
2. The authors try hard to categorize the advanced LTSF models into four groups based on their core techniques.
3. The Figures are simple and easy to understand, effectively presenting the model's design concept intuitively.

**Weaknesses:**

Generally, the technique of the manuscript is partially sound, and some major concerns are listed below:

1. The motivation of this paper is unclear, as noted in question 1. There is no evidence provided to demonstrate that combining effective components from related work will result in a better model. Even in the ablation study, the authors do not discuss this problem. Authors can analysis  why combining these components should lead to improved performance.
2. The authors review and categorize existing methods for long-term time series forecasting, stating that their classification is based on key techniques. However, the four categories are not on the same level. For instance, "Generalized Linear Dependency" would more accurately correspond to methods addressing nonlinear dependencies, representing a classification by learning framework. Yet, the authors unexpectedly group this with specific modeling techniques that focus on better leveraging time series characteristics within the same category.  Please provide a clearer justification for why you grouped these different types of approaches together, if you choose to keep the current categorization.
3. The authors should provide explanations for the function of symbols nearby, especially for less common symbols, such as $\times_2$ in Equation (2), even though they are defined in the appendix.

**Questions:**

1. The classification criteria in Figure 1 are unclear. And what are the advantages of each category, and what are the benefits of combining these advantages? I believe this information is essential for the paper. Additionally, in the overlapping section on the right of the figure, does the “…” indicate a lack of relevant work, or that the authors couldn’t find any? If none exist, why is there no related work in this area?
2. In section 3.1, the encoding part includes a series of transformer-based blocks, and the decoder operates with a series of MLPs. Could this increase the model's complexity?
3. Can you further explain “to maintain the same volume of information while shortening the length of each dimension, thereby reducing the computational cost of self-attention.” in line 237?

---

> ### Author Response · Authors · 2024-11-21
>
> Dear Reviewer 5FaK,
>
> Many thanks for identifying our work's soundness and providing useful comments and suggestions. Here are our answers to your concerns and questions:
>
> **Q1 & W1: About the motivation of this paper and the classification criteria of Figure 1**
>
> Thank you for your insightful questions and valuable suggestions. We sincerely appreciate your careful reading and the opportunity to clarify several points.
>
> We fully understand your concern about the "unclear motivation" behind our model. Upon reflection, we realize that we did not provide enough explanation about the characteristics and advantages of these techniques that are required to fully appreciate our analysis. To address this, we have added more details in the **Recent Advancements in LTSF** section of the Related Work part. We hope this addition will help potential readers better approve the motivation of our approach.
>
> Additionally, we are grateful for the reviewer’s reminder regarding the term "Generalized Linear Dependency." Initially, we used "Linear Dependency" to refer to forecasters that generate forecasting results as linear combinations of time tokens (components, bases). However, since models based on MLPs are not purely linear due to the presence of nonlinearities, we opted for the more accurate term, **Generalized Linear Projection** to avoid any ambiguity.
>
> To clarify the motivation behind our design, we would like to address the following points:
>
> 1. **Multi-scale modeling and time series decomposition** have been widely explored with complex techniques and are proven to be beneficial for capturing both coarse and fine-grained features of time series data. We integrate these concepts as multi-level learning in our framework.
> 2. Models like DLinear and N-BEATS, which use linear or MLP layers as their backbone, primarily focus on capturing linear dependencies in time series. These models make linear combinations of time tokens, forming what we refer to as "time bases."
> 3. Time series data inherently contains **redundant information**, which hinders dependency capturing and limits the utilization of long-term history. In our opinion, models like **Informer and PatchTST** have addressed this issue by using different methods to reduce information redundancy. Moreover, **LTSF models with frequency analysis capabilities, like FEDFormer and FITS**, deal with this problem by removing low-frequency components, which also contributes to the technique group we term "information compression."
> 4. A key debate in the LTSF community revolves around the effectiveness of **transformers versus linear models** as the primary forecasters. This has inspired further exploration into the best ways to capture dependencies in time series data, especially across different tokens such as time tokens and variate tokens.
>
> With this context, we categorize modern techniques for LTSF into four groups based on the primary techniques they utilize. While our classification is somewhat subjective, these techniques have been extensively validated, either theoretically (e.g., RLinear [1], N-Hits, Informer, FEDFormer) or experimentally (e.g., PatchTST, DLinear, iTransformer). We combine them into a unified framework because they are technically independent, and we believe integrating them provides a comprehensive solution for long-term time series forecasting.
>
> As noted by the reviewer Nedg, "Indeed, integrating various current methods is a good direction, and the methods used are logically reasonable." We agree with this sentiment but also recognize your concern that combining these methods may not always lead to incremental improvements. This is reflected in our final results, where we show that the individual components are useful in different scenarios, leading to a versatile forecaster capable of handling a wide range of forecasting tasks.
>
> In particular, we integrate the following core techniques into our model:
>
> - **MoMSA** for handling "various attention tokens"
> - **Residual Information Back** for addressing "information redundancy"
> - **Additional dimensions in the 3D time series** for capturing "multi-level features"
> - **MLP decoders** for modeling "generalized linear dependencies $\rightarrow$ generalized linear projection"
>
> We provide ablation studies on these techniques to demonstrate their individual contributions. Furthermore, we highlight the importance of each component through comparisons with previous models, e.g.,  **Informer** and **FEDformer** ( lines 407-409 in the revised manuscript), and so forth. These experiments validate the effectiveness of redundancy reduction in improving forecasting performance.
>
> [1] Li, Zhe, et al. "Revisiting long-term time series forecasting: An investigation on linear mapping." arXiv preprint arXiv:2305.10721 (2023).

---

> ### Author Response · Authors · 2024-11-21
>
> Regarding the reviewer’s question about the "..." in the overlapping section on the right of the figure, we acknowledge the speculation that there may be no prior research addressing both the attention token problem and information compression for LTSF. This is partly due to the recent emergence of the attention token problem, notably discussed by **iTransformer** last year. This breakthrough has yet to be fully integrated with information compression techniques in LTSF.
>
> We hope these clarifications help resolve the concerns raised and provide a clearer understanding of the motivation and rationale behind our design.

---

> ### Author Response · Authors · 2024-11-21
>
> **W2: "clearer justification for why you grouped these different types of approaches together"**
>
> First of all, we apologize for the typos and the incorrect names in some of the figures, and we have corrected some potential issues in the revised manuscript.
>
> Regarding the classification criteria presented in **Fig. 1**, We would like to provide the following explanations on the motivation behind this figure:
>
> 1. **Originality of the Figure**: Our intent in creating this figure was to summarize the most significant and breakthrough works in the LTSF field over the past five years. While each work presents a variety of components, we noticed that certain key techniques consistently emerge across multiple studies as fundamental contributors to improving LTSF performance. These recurring techniques formed the basis of our categorization.
> 2. **Techniques Chosen for Categorization**: We acknowledge that, as the reviewer pointed out, these techniques may not all be on the same level. However, our reasoning is that each of these factors significantly influences LTSF performance in an independent manner. Specifically, each technique was introduced in foundational papers, played a pivotal role in advancing the field, and, even though their forms may have evolved over time, their core ideas have remained consistent. Over the years, these ideas have been refined, modified, and demonstrated in various ways, but they continue to be central to state-of-the-art models. We believe that integrating these well-established techniques into a single framework is both meaningful and feasible, which is why we chose to highlight them.
> 3. **Methods within Each Group**: The methods listed in the figure are meant to serve as representative examples. We are aware that many other excellent methods could also be categorized within these groups. Due to space constraints, we selected a few examples from the state-of-the-art methods. Our approach was to categorize methods based on the key technique they predominantly rely on, though we recognize that many methods may incorporate elements from multiple groups.
>
> We hope this explanation clarifies the motivation behind the categorization in **Fig. 1** and the reasoning behind our selection of techniques. We believe these core ideas remain relevant and impactful, and we aim to integrate them into a unified framework that can further advance LTSF research.
> ****
> **W3: "Provide explanations for the function of symbols nearby"**
> Thank you for your kind reminder. We have followed your suggestion and made improvements to enhance the readability of the manuscript.

---

> ### Author Response · Authors · 2024-11-21
>
> **Q2: About the complexity of using transformers as the encoder and MLPs as the decoder.**
>
> The asymmetric architecture of TimeCapsule is a deliberate and carefully considered design. As mentioned in the introduction (lines 84–92), the motivation behind this choice is clear:
>
> - We use weak transformers as the encoder because we need their ability to capture multimode dependencies. This enables the model to extract relevant features and compress information effectively, which can then be used by the decoder for linear predictions.
> - We use MLPs as the decoder because we believe that the key to Long-Term Time Series Forecasting (LTSF) lies in capturing generalized linear dependencies, as discussed in the experimental section.
> - The use of this asymmetric architecture allows us to explore the respective roles of transformers and linear models in LTSF, striking a balance between the two approaches.
>
> Regarding the efficiency of this architecture, the cost of self-attention is significantly reduced through our MoMSA approach. As demonstrated in the **Efficiency Study** section, the FLOPs of TimeCapsule are reasonable compared to other state-of-the-art models. However, we acknowledge that further improvements in time and space efficiency are areas we plan to address in future research.
> ****
> **Q3: About the statement "to maintain the same volume of information while shortening the length of each dimension, thereby reducing the computational cost of self-attention"**
>
> Thank you for your thoughtful question. We're happy to provide a more detailed explanation.
>
> The main challenge with self-attention is its squared time complexity relative to the length of the dimension it operates on. One of the key advantages of our MoMSA approach is that it applies self-attention to a compressed dimension, significantly reducing the time complexity of transformers. This is also why we refer to it as "weak-transformers" — it reduces the computational burden while maintaining the core functionality of attention.
>
> Naturally, compressing the dimensions may raise concerns about potential information loss. However, we have implemented several measures to mitigate this risk. To clarify further, let's frame this from the view of information: If we consider a time series with 400 timestamps and 250 variables as an information carrier, the total information volume is (informally) $400 \times 250 = 100,000$. By transforming this 2D structure into a 3D container, we can represent the same amount of information as $100,000 = 25 \times 160 \times 25$. This reduces the dimensional complexity, allowing self-attention to operate on these reduced dimensions, which significantly lowers the computational cost compared to applying self-attention to the original $400 \times 250$ matrix. From another view, the MoMSA moves the contained information to the attention tokens by increasing the length of each, instead of increasing the number of attention tokens which may result in a major computation burden.
>
> In summary, this approach strikes a balance between reducing complexity and maintaining the essential information, while also ensuring that computational resources are used more efficiently.
>
> Authors

---

> ### Author Response · Authors · 2024-11-25
> **Rebuttal Deadline Approaching**
>
> Dear Reviewer 5FaK,
>
> We've answered all your questions and made modifications based on your suggestions. We sincerely appreciate your thoughtful questions and suggestions, which have taken our manuscript to the next level. We hope we've addressed all your concerns, and we'd love to hear your feedback. Please let us know if you have any further questions.
>
> Yours sincerely,
>
> The Authors

---

> > ### Comment · Reviewer_5FaK · 2024-11-26
> >
> > Thank you for the thorough response. I acknowledge that the four components mentioned by the authors are indeed important for the development of LTSF **individually**. However, based on Occam's Razor, we believe that integrating these components may not necessarily add value. This is reflected in the paper, where combining the four parts leads to a model with high memory usage, as shown in Figure 4.
> >
> > Based on this understanding, I decide to maintain my current rating.

---

> ### Author Response · Authors · 2024-11-26
> **Thanks for your reply !**
>
> Thank you for your feedback.
>
> We appreciate the reviewer's opinion, but we respectfully disagree. Firstly, memory usage has no direct relationship to the components or designs considered in this research. TimeCapsule incorporates the four ideas in a streamlined way, each corresponds to at most one minor design in the model. Memory usage depends mainly on normal model configurations, such as the self-attention dimension and the linear projection dimension, which can be significantly reduced in practice with little loss of performance, as explained in the manuscript.
>
> Second, We believe that value should not be judged solely on the basis of model performance. Not every ingredient should contribute to performance improvement. LTFS is a comprehensive task that should take many factors into account. For example, regarding the long-context bottleneck, by using dimension compression, we can achieve an alternative way to realize satisfactory long-history information utilization compared to patching.
>
> The only key design dedicated to improving the prediction performance may be the MoMSA, which combines dimension compression, multi-level learning, and multi-mode dependency capturing (different attention tokens). This is a compact and efficient module rather than a simple composition. In fact, what TimeCapsule has achieved is obtaining all of these properties in an effective yet simple way, which is in line with Occam's Razor, as mentioned by the reviewer.
>
> From this point of view, we really hope the reviewer can see why we're doing this research; and we would appreciate it if the reviewer could reconsider his judgment and decision.

---

### Official Review · Reviewer_CdzQ · 2024-11-02

**Soundness:** 2
**Presentation:** 2
**Contribution:** 3
**Rating:** 6
**Confidence:** 4

**Summary:**

This paper proposes TimeCapsule, a new architecture for time-series forecasting by combining key model-blocks introduced in recent works. Additionally, this paper employs a self-supervised loss based on JEPA to enhance the forecasting performance. Overall, TimeCapsule consists of an encoder to learn representations that are then used for forecasting with an MLP-based decoder. The key model-components seem to be: multi-level modelling, multi-mode dependency, and compressed representation forecasting.

**Strengths:**

* The paper presentation is very good considering that the model architecture being proposed in the paper is rather complex. Fig. 2 was also very helpful in understanding the various model components.
* It is interesting to see the JEPA loss being applied on time-series.
* The performance of the new architecture is evaluated on long-term time series forecasting datasets.
* The paper presents some ablations and also analyses the compute-efficiency.

**Weaknesses:**

* The main weakness, in my opinion, is that the paper only considers time-series forecasting to evaluate the model. To establish generality of the model architecture, it may be important to also consider other tasks such as classification, self-supervised learning, or irregular time-series forecasting/imputation.
* The paper makes many contributions and it may be more valuable to carefully ablate each of these new components. Some of the design decisions/ alternatives could be clearly elaborated. For example, why apply EMA over input time-series Y in Eq. 15? Why introduce Gaussian noise at the start of the block (L. 251) --- what alternatives can we consider in practice?
* The results suggest that some gains by JEPA loss are marginal. It'd have been nice if the authors demonstrated self-supervised pretraining based on JEPA helped various downstream tasks (e.g., forecasting/classification/etc).

Overall, I feel that this architecture is interesting and can be valuable for future time-series applications. However, due to the introduction of many new architectural blocks (not necessarily a bad thing), the performance on other time-series applications is not clear. It would help to at least demonstrate self-supervised pretraining based on JEPA for downstream applications; this may be compared against self-supervised learning by Patch-TST (e.g., JEPA/masked-pretraining).

**Questions:**

1. Based on your experiments, is TimeCapsule architecture key for the JEPA training? For example, if you tried to apply JEPA to PatchTST architecture, would we notice any improvements?
2. In line 368, it says "Besides, our model outperforms PatchTST, particularly on larger datasets, without employing patching techniques." Why is it beneficial to eliminate patching?
3. What kind of attention is used in the Tunnel layers?
4. What is coefficient considered for JEPA loss?

Minor typo: Both MvTS and MTS seem to refer to multivariate time series and it may be better to use just one abbreviation.

---

> ### Author Response · Authors · 2024-11-21
>
> Dear Reviewer CdzQ,
>
> Many thanks for identifying our work's good representations and providing useful comments and suggestions. Here are our answers to your concerns and questions:
>
> **W1: The main weakness, in my opinion, is that the paper only considers time-series forecasting to evaluate the model. To establish the generality of the model architecture, it may be important to also consider other tasks such as classification, self-supervised learning, or irregular time-series forecasting/imputation.**
>
> We greatly appreciate Reviewer CdzQ's thoughtful consideration. We completely agree that extending the proposed methods to self-supervised learning and other relevant tasks is a promising direction for future research.
>
> First and foremost, we have made efforts to design experiments implementing JEPA-based pre-training; however, the initial results were not as encouraging as we had hoped. These results, along with our analysis of potential reasons, have been documented in **Appendix G** for reference. We hope these insights will be valuable for future explorations in this area.

---

> > ### Author Response · Authors · 2024-11-21
> > **Table for classification**
> >
> > | Datasets              | Informer (2021) | Pyraformer (2021) | Autoformer (2021) | FEDformer (2022) | iTransformer (2023) | Dlinear (2023) | TiDE (2023) | Timesnet (2022) | TimeCapsule |
> > | --------------------- | ---------------- | ----------------- | ----------------- | ----------------- | ------------------- | -------------- | ----------- | --------------- | ----------- |
> > | **Heartbeat**          | 80.5             | 75.6              | 74.6              | 73.7              | 75.6                | 75.1           | 74.6        | 78.0            | 78.5        |
> > | **FaceDetection**      | 67.0             | 65.7              | 68.4              | 66.0              | 66.3                | 68.0           | 65.3        | 68.6            | 70.2        |
> > | **Handwriting**        | 32.8             | 29.4              | 36.7              | 28.0              | 24.2                | 27.0           | 23.2        | 32.1            | 27.0        |
> > | **SelfRegulationSCP2** | 53.3             | 53.3              | 50.6              | 54.4              | 54.4                | 50.5           | 53.4        | 57.2            | 57.8        |
> > | **EthanolConcentration** | 31.6           | 30.8              | 31.6              | 28.1              | 28.1                | 32.6           | 27.1        | 35.7            | 32.0        |
> > | **UWaveGestureLibrary** | 85.6            | 83.4              | 85.9              | 85.3              | 85.9                | 82.1           | 84.9        | 85.3            | 88.8        |
> > | **Average Accuracy**   | 58.5             | 56.4              | 58.0              | 55.9              | 55.8                | 55.9           | 54.8        | 59.5            | 59.1        |

---

> > ### Author Response · Authors · 2024-11-21
> >
> > Additionally, the classification performance of TimeCapsule is somewhat beyond our initial expectations. Because all of the proposed designs in this study were motivated by key ingredients identified from prior work in LTSF and tailored specifically to address challenges in this area. As such, we acknowledge the reviewer's concern that this initial version of TimeCapsule may not be directly applicable to other tasks, such as classification, without suitable modifications. For example, the alignment we achieve via JEPA centers on historical and future information for predictive representation, which may not directly support tasks like classification. To broaden TimeCapsule’s applicability, adjustments such as rethinking the use of weak transformers for encoding or MLPs for decoding would likely be necessary while preserving the core structure of the framework.
> >
> > Lastly, we do not want to distract the model’s focus and purpose, as outlined in the title: **Multivariate Long-Term Time Series Forecasting.** Our primary aim in this research has been to study and enhance **predictive representations**, a concept central to forecasting. As a novel framework, TimeCapsule is not yet intended to be a universal solution for all time-series tasks. Forecasting itself is a long-standing and multifaceted challenge, and we chose it as a starting point for exploration. In fact, the **versatility** we emphasize in this work primarily pertains to the model’s ability to handle diverse forecasting scenarios and dataset characteristics effectively.
> >
> > Overall, we would like to sincerely thank Reviewer CdzQ for his attention to the generality of TimeCapsule. We would like to emphasize that rather than solely introducing a new model and benchmarks, this paper aims to revisit and address the challenges of LTSF, offering fresh insights into the importance of various components for effective forecasting. Through experiments and detailed discussions, we hope to contribute not only a novel framework but also a deeper understanding of the field.

---

> ### Author Response · Authors · 2024-11-21
> **Table for JEPA pre-training**
>
> | Models                                                         | Metric | **TimeCapsule (fine-tune)** MSE | **TimeCapsule (fine-tune)** MAE | **TimeCapsule (supervised)** MSE | **TimeCapsule (supervised)** MAE | iTransformer MSE | iTransformer MAE | TimeMixer MSE | TimeMixer MAE | **PatchTST (self-supervised)** MSE | **PatchTST (self-supervised)** MAE | Crossformer MSE | Crossformer MAE |
> | -------------------------------------------------------------- | ------ | ------------------------------- | ------------------------------- | ------------------------------- | ------------------------------- | ---------------- | ---------------- | ------------- | ------------- | ---------------------------------- | --------------------------------- | --------------- | --------------- |
> | **Weather**   | 96     | _0.142_  | _0.187_     | **0.141**   | **0.186**    | 0.159 | 0.208   | 0.147 | 0.198  | 0.148  | 0.196   | 0.146           | 0.212           |
> |                                                                | 192    | _0.188_    | _0.233_  | **0.187**    | **0.232**    | 0.200  | 0.248            | 0.192         | 0.243         | 0.193                              | 0.240                             | 0.195           | 0.261           |
> |                                                                | 336    | **0.236**                       | **0.271**                       | _0.239_                         | _0.272_                         | 0.253            | 0.289            | 0.247         | 0.284         | 0.244                              | 0.279                             | 0.268           | 0.325           |
> |                                                                | 720    | _0.310_ | _0.325_   | **0.309**                       | **0.323**                       | 0.321            | 0.338            | 0.318         | 0.330         | 0.321                              | 0.334                             | 0.330           | 0.380           |
> | **Traffic**                                                     | 96     | 0.369                           | _0.252_                         | **0.361**                       | **0.246**                       | _0.363_          | 0.265            | 0.466         | 0.294         | 0.382                              | 0.262                             | 0.514           | 0.282           |
> |                                                                | 192    | 0.388    | _0.260_    | **0.383**      | **0.257**                       | _0.385_          | 0.273            | 0.508         | 0.299         | _0.385_                              | 0.261                             | 0.501           | 0.273           |
> |                                                                | 336    | _0.396_    | _0.265_                         | **0.393**                       | **0.262**                       | _0.396_          | 0.277            | 0.526         | 0.309         | 0.409                              | 0.275                             | 0.507           | 0.278           |
> |                                                                | 720    | _0.435_                         | _0.288_                         | **0.430**                       | **0.282**                       | 0.445            | 0.312            | 0.554         | 0.322         | 0.438                              | 0.291                             | 0.571           | 0.301           |
> | **Electricity**                                                 | 96     | _0.127_                         | _0.221_                         | **0.125**                       | **0.218**                       | 0.138            | 0.237            | 0.131         | 0.224         | 0.132                              | 0.227                             | 0.135           | 0.237           |
> |                                                                | 192    | _0.146_                         | _0.239_                         | _0.146_                         | **0.238**                       | 0.157            | 0.256            | 0.151         | 0.242         | 0.148                              | 0.241                             | 0.160           | 0.262           |
> |                                                                | 336    | _0.163_                         | _0.256_                         | **0.161**                       | **0.254**                       | 0.167            | 0.264            | 0.169         | 0.260         | 0.167                              | 0.260                             | 0.182           | 0.282           |
> |                                                                | 720    | 0.198                           | 0.287                           | _0.194_                         | **0.285**    | _0.194_          | _0.286_          | 0.227         | 0.312         | 0.205  | 0.292         | 0.246           | 0.337           |

---

> ### Author Response · Authors · 2024-11-21
>
> **W2: The paper makes many contributions and it may be more valuable to carefully ablate each of these new components. Some of the design decisions/ alternatives could be clearly elaborated. For example, why apply EMA over input time-series Y in Eq. 15? Why introduce Gaussian noise at the start of the block (L. 251) --- what alternatives can we consider in practice?**
>
> We fully agree with the reviewer’s insights, and we have made concerted efforts to investigate the effects of the various components in our model, as described in the Model Analysis section. Specifically, we have explored the impact of the main components outlined in Section 3.2:
>
> 1. **Effects of Residual Information Compensation**: This refers to the **Residual Information Back** mechanism, where we investigate the influence of residual information on forecasting performance.
> 2. **Effects of JEPA and Predictive Representation**: This analysis pertains to the **Representation Prediction with JEPA Loss**, evaluating how the model benefits from using JEPA to generate predictive representations.
> 3. **Analysis of Asymmetric Structure**: This addresses the dimensionality reduction process, which partially reflects the multimodal dependencies captured by **MoMSA**.
>
> In addition to the core components mentioned above, we’d like to clarify the rationale behind some of the additional design choices:
>
> - **Why Apply EMA over Series Y?** The motivation for this choice is detailed in lines 291–293. We aim to address the inconsistency in sequence length between the input and output, particularly when the length of **Y** exceeds that of **X**. In such cases, we cannot directly apply the encoder to transform **Y** into a compressed representation, since the encoder is fixed to the input length. To solve this issue, we apply **EMA** to smooth and adjust the length of **Y** to match **X**. The rationale behind using EMA is based on the assumption that, generally, the correlations between nearby timestamps are stronger. We see the whole series as an *information carrier*, and convolve it to integrate the contained information as the purpose of JEPA is to align the *information* contained in the representations between history and the future. In this context, EMA serves as an efficient method for integrating information in the series. We have added a short explanation of this choice in the new manuscript (lines 309~311) for clarity.
> - **Why Add Gaussian Noise?** The decision to add Gaussian noise is based on two considerations:
>   1. From an information theory and coding theory perspective, noise can serve as a form of redundancy that helps improve robustness during compression (*channel coding*). This aligns with our goal of minimizing information loss in the encoding stage (lines 258~263, new pdf). Although we initially assumed this was beneficial, we have conducted a new ablation study (included in **Appendix D.2**), which shows that adding noise provides modest benefits.
>   2. There is also an empirical basis for this decision. In deep learning, adding controlled oscillations (such as noise) during training has been observed to improve the generalization ability of models. While this effect is relatively small, it can be seen as a simple technique to enhance robustness. We agree with the reviewer’s thoughtful suggestion, and we have added an additional ablation study about this in the manuscript **Appendix**.
>
> Finally, regarding potential alternatives in practice: Instead of using EMA, one could simply truncate **Y** to match the length of **X**, given that it is not necessary to include all the information within the forecasting target for the predictive representation learning. Another option could be to increase the input window size or adjust the forecast horizon, both of which are practical alternatives for real-world applications. Since the noise introduced in the encoder sometimes shows slight effectiveness, it may be ok to remove it in practice when the time series dataset is large and robust enough.
> ****
> **W3 The results suggest that some gains by JEPA loss are marginal. It'd have been nice if the authors demonstrated self-supervised pretraining based on JEPA helped various downstream tasks (e.g., forecasting/classification/etc).:**
>
> please see the results shown in W1.

---

> > ### Author Response · Authors · 2024-11-21
> > **Results of ablation on the introduced Gaussian noise**
> >
> > | **Method** | **ETTm2** (MSE avg) | **ETTm2** (MAE avg) | **Weather** (MSE avg) | **Weather** (MAE avg) | **Traffic** (MSE avg) | **Traffic** (MAE avg) | **Electricity** (MSE avg) | **Electricity** (MAE avg) |
> > |------------|---------------------|---------------------|-----------------------|-----------------------|-----------------------|-----------------------|---------------------------|---------------------------|
> > | **Origin** | 0.247               | 0.308               | 0.219                 | 0.253                 | 0.392                 | 0.262                 | 0.157                     | 0.249                     |
> > | **w/o noise** | 0.250             | 0.310               | 0.224                 | 0.258                 | 0.392                 | 0.262                 | 0.219                     | 0.265                     |

---

> ### Author Response · Authors · 2024-11-21
>
> **Q1: Based on your experiments, is TimeCapsule architecture key for the JEPA training? For example, if you tried to apply JEPA to PatchTST architecture, would we notice any improvements?**
>
> Thank you for your insightful question regarding the transferability of JEPA utilization.
>
> We carefully considered this aspect when initially designing TimeCapsule and explored the possibility of applying JEPA beyond our specific architecture. In fact, it is not strictly necessary to use the TimeCapsule framework for applying JEPA. Any model that generates meaningful predictive representations in its hidden layers can compute this additional loss, as has been demonstrated in previous work [1]. Therefore, models like PatchTST could also potentially benefit from this technique by introducing an inductive bias into the learned representations, provided that the same encoder is used consistently to encode the relevant information for the JEPA loss computation.
>
> However, we acknowledge that the purpose of this computation requires careful consideration. Questions such as **What information needs to be aligned?** and **Why to align this information?** are essential to understanding the potential impact of JEPA. In our case, the motivation is clear and grounded in the specific requirements of time series forecasting. As discussed in the new manuscript (lines 294~299), we assume a natural gap between the present and the future when making predictions. To bridge this gap, we introduce a representation predictor and JEPA loss (as shown in Fig. 2) to transmit the compressed auxiliary information from the past to inform future predictions. The improvements brought by JEPA are not guaranteed and indeed vary across datasets, as the impact of JEPA depends on the specific characteristics of each dataset. We have discussed these findings and the evaluation of this assumption in the experiment section titled **Effects of JEPA and Predictive Representation.**
>
> Once again, thank you for raising such a thoughtful question. We believe that further exploration is required to determine the most effective and stable use of JEPA in time series forecasting, and we are excited to continue investigating this direction.
>
> [1] Alexei Baevski et al., Data2vec: A general framework for self-supervised learning in speech, vision, and language. In International Conference on Machine Learning, pp. 1298–1312. PMLR, 2022
> ****
> **Q2: In line 368, it says "Besides, our model outperforms PatchTST, particularly on larger datasets, without employing patching techniques." Why is it beneficial to eliminate patching?**
>
> Thank you for your careful reading and thoughtful comments. We regret that our explanation may have led to a misunderstanding. We have never suggested that eliminating batching would be beneficial. On the contrary, patching is indeed a useful and important technique in many areas, particularly for time series, as it helps reduce the computational complexity of self-attention and allows the model to better leverage long-term history information.
>
> However, we do recognize that patching has some inherent limitations, as discussed in the introduction (lines 66-72). This is precisely why we propose an alternative approach—dimension compression—to complement or replace patching. Our results show that our approach can outperform PatchTST, demonstrating that this technique is both useful and, in some cases, more effective.
> ****
> **Q3: What kind of attention is used in the Tunnel layers?**
>
> In our model, we use the original vanilla transformer blocks (i.e., standard linear dot-product self-attention, as described in the initial transformer paper ("Attention is all you need")) to act as the "tunnels" for processing each compressed dimension, as indicated in the caption of Fig. 2. These transformer blocks are applied to different compressed dimensions after the specific transformation block. Their purpose is to ensure the scalability of the model, improve dependency capturing, and help obtain deeper, more informative representations. This design choice may enhance the flexibility and capacity of our model to capture complex multi-mode and single-mode relationships.
> ****
> **Q4: What is the coefficient considered for JEPA loss?**
>
> Regarding the combination of losses. In our paper, we directly add the two losses without weighting them differently. We have experimented with other combinations of coefficients for the two losses, but the performance did not improve. However, based on the discussion in the experiment section and the preliminary exploration of JEPA pre-training as you have suggested, we believe it may be valuable to explore more sophisticated strategies for utilizing the JEPA loss, such as introducing dynamic weighting or adapting the loss combination throughout the training process. We look forward to investigating this further in future work.

---

> ### Author Response · Authors · 2024-11-21
> **Typos in the manuscript**
>
> We have modified all "MTS" to "MvTS", thank you for your kind reminder.

---

> > ### Comment · Reviewer_CdzQ · 2024-11-24
> > **Thank you for your experiments!**
> >
> > Thank you for the detailed rebuttal response and additional experiments. My concerns are mostly addressed. I request the authors to enhance the readability of section 3 with more information/intuition on the design decisions in this architecture. I'm currently raising the score to 6. Good luck!

---

> ### Author Response · Authors · 2024-11-24
> **Many thanks for your suggetions and raising score**
>
> Thank you so much for your response! We'd love to add more explanations and intuitions about our design. It's really an instructive suggestion to make the paper more understandable.
>
> Thanks again for your time and your valuable review.

---

### Official Review · Reviewer_MW4b · 2024-11-03

**Soundness:** 3
**Presentation:** 3
**Contribution:** 3
**Rating:** 6
**Confidence:** 3

**Summary:**

This manuscript introduces a framework that adds an extra level dimension to time series data and employs mode production to capture multi-mode dependencies. It utilizes transformer blocks to capture dependencies across temporal, level, and variate dimensions, respectively.

**Strengths:**

This manuscript is well-written, logically organized, and easy to follow. It provides a comprehensive analysis of recent SOTA methods for MTS forecasting. The Mode Specific Multi-head Self-attention utilizes a vanilla attention mechanism to capture temporal dependencies. Although patching has been a widely used technique in recent years to reduce the length of input for attention mechanisms, this manuscript employs low-rank transforms as a replacement, achieving promising efficiency.
TimeCapsule applied the JEPA approach to time series forecasting and demonstrates its effectiveness. TimeCapsule has achieved superior performance in both accuracy and computational speed. Extensive ablation studies were conducted, which underscore the effectiveness of the core components of TimeCapsule.
In my opinion, this manuscript is novel and likely to generate interest among researchers in the field of time series analysis.

**Weaknesses:**

Existing transformer-based methods partition time series sequences into patches of various sizes to capture multi-scale temporal dependencies. TimeCapsule introduces an additional level dimension to the 2-dimensional MTS data to capture multi-scale dependencies. However, the explanation of how multi-scale dependencies are captured is lacking. Specifically, definitions of how multi-scale levels and variate dependencies are defined are needed.

In Figure 2, the components below ReVin_Norm are depicted in a way that may be misleading. Consider using more intuitive shapes to represent tensors.

Table 2 presents an ablation study of the residual information connection. For ETTh2 and ETT m2 datasets, the accuracy achieved with original information is similar to that achieved with residual information connections. In addition, the gap between the two settings increases as the forecasting horizon extends. An analysis of these findings would be beneficial.

**Questions:**

I have a few suggestions that may enhance the quality of the paper:

(a) Please consider increasing the text size in some figures to improve readability.

(b) Could you explain the rationale behind connecting the T-TransBlock, L-TransBlock, and V-TransBlock in series? Have you tested configuring these blocks in parallel, concatenating their outputs, and then using a linear layer to generate predictions?

(c) Consider adding a description in Section 3.2 of how multi-scale dependencies for Temporal, Level, and Variate dimensions are captured.

---

> ### Author Response · Authors · 2024-11-21
>
> Dear Reviewer MW4b:
>
> Thank you for identifying the novelty of our work, based on your suggestions, we believe the quality of this paper has been significantly improved.  Here are our answers to your concerns and questions:
>
> **W1 & Q3: About the meaning of multi-level and the sequence structure of TimeCapsule**
>
> Thank you for your careful reading and valuable suggestions.
>
> In this manuscript, the term *multi-level* refers to an integrated concept encompassing both multi-scale modeling and time series decomposition. In general, time series data can benefit from multi-frequency modeling and decomposition into different components or bases, allowing both coarse and fine-grained features to be captured simultaneously. To facilitate this, we introduce a level dimension that enables the model to learn these properties independently in a simple yet effective way. Experimental results and subsequent analyses demonstrate that our model successfully captures these multi-level dependencies.
>
> In response to your suggestions, we have added a concise explanation of how temporal, level, and variate dimensions are addressed in the final paragraph of **Section 3.2**. Additionally, we have further investigated this ability in **Appendix E**, where we explore in detail the mechanism of what has been learned by TimeCapsule in the way of answering three questions, many interesting observations and analyses have been added to the paper through visualization. We hope these clarifications and additional experiments strengthen the interpretations and offer a more comprehensive understanding of our approach.
> ****
> **W2:  the components below ReVin_Norm are depicted in a way that may be misleading**
>
> Thank you for pointing that out. We have modified and clarified certain aspects of Fig. 2 to make the meaning more clear. We hope that these changes address your concerns and provide the clarity you were expecting.
> ****
>  **W3: Table 2 presents an ablation study of the residual information connection. For ETTh2 and ETT m2 datasets, the accuracy achieved with original information is similar to that achieved with residual information connections. In addition, the gap between the two settings increases as the forecasting horizon extends. An analysis of these findings would be beneficial.**
>
> Thank you for your careful attention to this point. We agree that this finding could inspire more valuable insights for LTSF. In response, we have added a few analyses (thoughts) regarding this phenomenon (please see lines 409 ~ 415 in the revised manuscript).
> ****
>  **Q1: Please consider increasing the text size in some figures to improve readability.**
>
>  Thank you for your kind reminder.  We have increased the text size in several figures to improve readability.

---

> ### Author Response · Authors · 2024-11-21
>
> **Q2: Could you explain the rationale behind connecting the T-TransBlock, L-TransBlock, and V-TransBlock in series? Have you tested configuring these blocks in parallel, concatenating their outputs, and then using a linear layer to generate predictions?**
>
> Thank you for your insightful question. This is indeed a key consideration in the design of our model. We chose a sequential structure over a parallel one for the following reasons:
>
> - **Inspiration from Compression Techniques:** Our model is inspired by the "bits-back coding" method from the compression domain, which achieves effective compression through sequential steps. As mentioned in the manuscript (lines 89~90), "Therefore, we propose TimeCapsule, a novel model employing a Chaining Bits Back with Asymmetric Numeral Systems (BB-ANS) -like architecture..." This is the first reason for adopting a sequential structure.
> - **Effectiveness of Sequential Compression:** Intuitively, compressing along each dimension sequentially is necessary to achieve deep compression. Parallel compression would not allow thorough reduction of redundancies within each dimension, as each compression step could be disturbed by redundancies from other dimensions, leading to inefficient compression and poorer information utilization.
> - **Scalability and Efficiency:** Implementing a parallel model would make it difficult to scale effectively and capture abstract representations. If we were to use parallel processing, each dimension would require an independent deep stack, resulting in prohibitive computational costs.
>
> While we believe the sequential design is more suitable for the reasons outlined, we acknowledge the potential value of exploring a parallel design in future work. We hope this explanation clarifies the reasoning behind our choice.
>
> Lastly, we would again like to sincerely thank Reviewer MW4b for recognizing the novelty and contribution of our work, and for providing valuable suggestions. We have made major revisions following all the reviewer's feedback, incorporating additional explanations, experiments, and analyses. While we hope the revisions address the concerns raised and improve the clarity of our work, we acknowledge that there is always room for further refinement. We truly appreciate the opportunity to revise our manuscript and hope it now is a more qualified research and can make a meaningful contribution to the field.
>
> Authors

---

> > ### Comment · Reviewer_MW4b · 2024-11-26
> >
> > I appreciate the authors' efforts in responding to my questions.

---

> ### Author Response · Authors · 2024-11-25
> **Rebuttal Dealine Approaching**
>
> Dear Reviewer MW4b,
>
> We've answered all your questions and made modifications based on your suggestions. We sincerely appreciate your thoughtful questions and suggestions, which have taken our manuscript to the next level. We hope we've addressed all your concerns, and we'd love to hear your feedback. Please let us know if you have any further questions.
>
> Yours sincerely,
>
> The Authors

---

> ### Author Response · Authors · 2024-11-26
> **Thanks for your reply !**
>
> Dear Reviewer MW4b,
>
> So glad to hear from you, and thanks so much for your appreciation of our work. Your suggestions have helped to enhance the quality of the manuscript. Following your questions and suggestions, we have made efforts to
>
> - enhance the readability of the text and figures.
>
> - add descriptions in **Section 3**, to help understand the structure design of TimeCapsule.
>
> - add new analyses to the empirical findings in the experiment part.
>
> - provide a series of visualizations in **Appendix E**, to explore and demonstrate the underlying mechanism of TimeCapsule in more detail.
>
> We would be really grateful if you could spare a little time to consider raising the score before the rebuttal deadline, which would be an exciting and crucial support to this borderline manuscript.
>
> Again, we sincerely appreciate your time and every question/suggestion you have posed.
>
> Wishing you all the best,
>
> The Authors.

---

### Author Response · Authors · 2024-11-26
**Summary of Revisions**

We would like to express our gratitude to all the reviewers for their insightful feedback and constructive comments, which have significantly improved the quality and completeness of our paper.

In this work, we propose a novel framework for Long-term Time Series Forecasting (LTSF) by considering four key ideas. Unlike prior approaches, we design and integrate these elements in a streamlined deep-learning architecture, ensuring both effectiveness and efficiency. We have evaluated our model in a comprehensive and challenging benchmark, demonstrating its advantages in terms of both performance and computational efficiency. Additionally, we conducted experiments to validate the effectiveness of each model component, while exploring interesting problems in the field through the proposed framework.

Overall, TimeCapsule is a novel, effective, and versatile model with a reasonable computational cost for LTSF. Through this framework, one can investigate many potential ingredients related to LTSF, as partially shown in this research.

We feel pleased that the reviewers have acknowledged the novelty, contributions, and presentations of our work:

- **Reviewer MW4b**: "Well-written, logically organized, and easy to follow"; "Novel and likely to generate interest among researchers in the field of time series analysis."
- **Reviewer CdzQ**: "The paper presentation is very good, considering the model architecture is complex"; "This architecture is interesting and can be valuable for future time-series applications"; "It is interesting to see JEPA loss applied to time series."
- **Reviewer 5FaK**: "Provides new insights into treating time series as a 3D tensor"; "The technique of the manuscript is partially sound."
- **Reviewer Nedg**: "The research direction is relatively innovative"; "Integrating various current methods is a good direction, and the methods are logically reasonable"; "The article’s expression and illustrations are overall very clear."

The reviewers also raised valuable and constructive concerns. In response, we have made the following revisions:

- **Typos** (All Reviewers): We carefully reviewed the manuscript for any typographical errors and resolved them.

- **Motivation and Model Design** (All Reviewers): We strengthened the explanation of our motivation in $\underline{\text{Section 1, 2}}$. Also, we clarified several minor design choices within the model in $\underline{\text{Section 3}}$, with further explanations and empirical results.

- **Interpretability and Further Exploration** (Reviewers MW4b, Nedg): We expanded the explanations on how multi-mode dependencies and multi-level properties are captured in  $\underline{\text{Section 3}}$. We also added a detailed visual exploration of this capability in  $\underline{\text{Appendix E}}$, which uncovered several interesting new insights. Additional analyses of these findings are included in  $\underline{\text{Section 4}}$ and  $\underline{\text{Appendix D}}$.

- **Generality** (Reviewer CdzQ): To broaden the scope of the model's applicability, we explored TimeCapsule’s potential for self-supervised learning with JEPA, as well as its performance in classification tasks, in $\underline{\text{Appendix G}}$.

All updates are highlighted in blue. We hope our response has fulfilled the reviewer's expectations and would be open to answering any further questions.

---

### Note · Authors · 2025-01-23

I have read and agree with the venue's withdrawal policy on behalf of myself and my co-authors.